# Over-squashing in
# Spatiotemporal Graph Neural Networks

**Ivan Marisca**[1]*     **Jacob Bamberger**[2]     **Cesare Alippi**[1,3]     **Michael M. Bronstein**[2,4]

[1]Università della Svizzera italiana, IDSIA, Lugano, Switzerland.
[2]University of Oxford, Oxford, UK.
[3]Politecnico di Milano, Milan, Italy.
[4]AITHYRA, Vienna, Austria.

## Abstract

Graph Neural Networks (GNNs) have achieved remarkable success across various domains. However, recent theoretical advances have identified fundamental limitations in their information propagation capabilities, such as over-squashing, where distant nodes fail to effectively exchange information. While extensively studied in static contexts, this issue remains unexplored in Spatiotemporal GNNs (STGNNs), which process sequences associated with graph nodes. Nonetheless, the temporal dimension amplifies this challenge by increasing the information that must be propagated. In this work, we formalize the spatiotemporal over-squashing problem and demonstrate its distinct characteristics compared to the static case. Our analysis reveals that, counterintuitively, convolutional STGNNs favor information propagation from points temporally distant rather than close in time. Moreover, we prove that architectures that follow either time-and-space or time-then-space processing paradigms are equally affected by this phenomenon, providing theoretical justification for computationally efficient implementations. We validate our findings on synthetic and real-world datasets, providing deeper insights into their operational dynamics and principled guidance for more effective designs.

## 1   Introduction

Graph deep learning [1, 2] has become a powerful paradigm for learning from relational data, particularly through graph neural networks (GNNs) [3–5]. These models process attributed graphs, where nodes represent entities and edges encode their relationships, and have shown strong performance in applications such as drug discovery [6, 7], material synthesis [8, 9], and social media analysis [10]. Beyond static graphs, GNNs have also been extended to dynamic settings where graph data evolves over time [11–13]. When data can be represented as synchronous sequences associated with nodes of a graph, a common approach is to pair GNNs with sequence models like recurrent neural networks (RNNs) [14, 15] or temporal convolutional networks (TCNs) [16, 17], leading to the class of spatiotemporal graph neural networks (STGNNs) [18–20]. These architectures have been successfully applied to real-world problems ranging from traffic forecasting [19, 21] to energy systems [22, 23] and epidemiology [24]. While the strong empirical performance of STGNNs has largely driven their development, much less attention has been devoted to understanding their theoretical capabilities and limitations. This contrasts with the literature on static GNNs, which includes extensive analyses on expressivity [25, 26] and training dynamics, including over-smoothing [27–29] and over-squashing [30–32]. In particular, over-squashing – where information from distant nodes is compressed through graph bottlenecks – has emerged as a central limitation of message-passing

---

*Work done while at University of Oxford. Correspondance to `ivan.marisca@usi.ch`

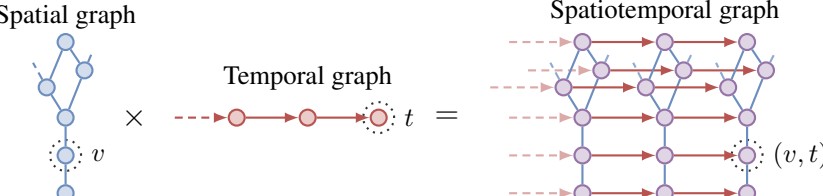

Figure 1: Example of spatiotemporal topology governing information propagation in STGNNs. The increasing receptive fields of graph-based and sequence-processing architectures compound, as shown in the Cartesian product of spatial and temporal graphs on the right.

architectures. However, these insights do not directly extend to STGNNs, in which information is propagated not only across the *graph* but also along the *time* axis. Understanding how over-squashing occurs in this joint spatiotemporal setting remains an open question.

The temporal axis represents an additional challenge, typically handled by imposing a locality assumption: adjacent time steps are presumed to be more correlated than distant ones. Under this assumption, the sequence behaves like a second, directed graph whose edges encode temporal order. As we stack local filters – or simply process longer sequences – increasing volumes of information are stored into fixed-width embeddings, a limitation already noted for recurrent architectures [33, 34]. When this temporal compression meets the constraints of spatial message passing, it creates a compound bottleneck we call *spatiotemporal over-squashing*: messages must cross rapidly expanding receptive fields in both space and time, exceeding the capacity of intermediate representations. Figure 1 visualizes this dual-graph structure, where information must propagate across both the spatial dimensions (blue edges) and temporal dimensions (red edges), with each path potentially contributing to the over-squashing phenomenon.

Two dominant architectural strategies have emerged to propagate information both through time and space. *Time-then-space* (TTS) models first compress each node's sequence into a vector representation and only afterward propagate these embeddings across the graph, whereas *time-and-space* (T&S) models interleave temporal and spatial processing so that information flows across both axes at every layer [35, 20]. While the choice between these paradigms has been driven mainly by empirical accuracy and computational costs, we argue that a principled analysis of these trade-offs is essential for guiding the design of future STGNNs.

In this work, we investigate how the interplay between temporal and spatial processing in STGNNs shapes learned representations, and how this process is limited by spatiotemporal over-squashing. We analyze information propagation patterns in existing STGNNs designs by tracing the sensitivity of each embedding to the input features contained at neighboring nodes and time steps. Specifically, our study focuses on convolutional STGNNs, whose temporal component is implemented through a shift operator that exchanges messages between adjacent time steps – the time-domain analogue of message-passing on graphs [36]. In summary, our contributions are the following:

1. We formally characterize spatiotemporal over-squashing and show its fundamental differences from the static case. The temporal dimension introduces an additional axis for information flow, potentially amplifying the compression effects observed in static graphs. (Sec. 3)

2. We prove both theoretically and empirically that architectures leveraging causal convolutions are, counterintuitively, more sensitive to information far apart in time, and we outline architectural modifications that mitigate this imbalance when required by the task. (Sec. 4)

3. We demonstrate that spatiotemporal over-squashing affects T&S and TTS paradigms to the same degree. Thus, the computational benefits of TTS models come at no extra cost in terms of information bottlenecks, providing theoretical support for scalable designs. (Sec. 5)

All theoretical findings are supported by empirical results on both synthetic tasks specifically designed to highlight spatiotemporal bottlenecks, and real-world benchmarks, demonstrating that our insights translate to practical improvements. To our knowledge, no previous work addressed the over-squashing phenomenon in STGNNs, despite its potential impact on spatiotemporal modeling. Our work fills this gap by providing a theoretical framework for understanding information propagation in STGNNs, with direct implications for model design and optimization.

## 2 Background

**Problem setting** We denote by $\mathcal{V}$ the set of $N$ synchronous and regularly-sampled time series, with $\boldsymbol{x}_t^v \in \mathbb{R}^{d_x}$ being the $d_x$-dimensional observation at time step $t$ associated with time series $v \in \mathcal{V}$. The sequence $\boldsymbol{x}_{t\text{-}T:t}^v \in \mathbb{R}^{T \times d_x}$ indicates the node observations in the interval $(t - T, t]$, $\boldsymbol{X}_t \in \mathbb{R}^{N \times d_x}$ the matrix of all observations in $\mathcal{V}$ at time $t$, so that $(\boldsymbol{X}_t)_v = \boldsymbol{x}_t^v$ is the $v$-th entry of $\boldsymbol{X}_t$. When referring to a generic node or time step, we omit the indices $\cdot{}^v$ and $\cdot{}_t$ if not required. We express temporal dependencies across observations within $\boldsymbol{x}_{t\text{-}T:t}$ as a directed path graph $\boldsymbol{T} \in \{0,1\}^{T \times T}$, named *temporal graph*, where $(\boldsymbol{T})_{ij}$, i.e., the edge from time step $t - i$ to $t - j$, is 1 only if $i - j = 1$; $\boldsymbol{T}$ acts as the *backward shift operator* [37]. We assume the existence of relationships across time series, describing dependencies or correlations between the associated observations. We express them as edges in a *spatial graph* with (weighted) adjacency matrix $\boldsymbol{A} \in \mathbb{R}_{\geq 0}^{N \times N}$, where $a^{uv} = (\boldsymbol{A})_{uv}$ is nonzero only if there exists an edge connecting node $u$ to $v$. We use $\widetilde{\boldsymbol{A}}$ to indicate a *graph shift operator*, i.e., an $N \times N$ real matrix with $\tilde{a}^{uv} \neq 0$ if and only if $a^{uv} \neq 0$ [38].

We focus on node-level tasks and, given a window of $T$ observations $\boldsymbol{X}_{t\text{-}T:t}$ and target label $\boldsymbol{Y}_t$, we consider families of (parametric) models $f_{\boldsymbol{\theta}}$ conditioned on the structural dependencies such that

$$\hat{\boldsymbol{y}}_t^v = (f_{\boldsymbol{\theta}}(\boldsymbol{X}_{t\text{-}T:t}, \boldsymbol{A}, \boldsymbol{T}))_v, \quad \forall\, v \in \mathcal{V}, \tag{1}$$

where $\boldsymbol{\theta}$ is the set of learnable parameters and $\hat{\boldsymbol{y}}_t^v$ is the estimate for $\boldsymbol{y}_t^v$. For classification tasks, $\boldsymbol{y}_t^v$ encodes the node label, while for prediction, the label is a sequence of $k$ future observations $\boldsymbol{x}_{t:t+k}^v$. Parameters are optimized using a task-dependent loss function, e.g., the mean squared error (MSE).

**Spatiotemporal message passing** STGNNs are architectures specifically designed to process graph-structured data whose node features evolve over discrete time steps [20, 13]. These models leverage GNNs to capture spatial dependencies while employing sequence-processing operators to model temporal dynamics. Among the different GNN variants, the primary deep learning approach for relational data are message-passing neural networks (MPNNs) [39], which operate by iteratively updating each node's representation through aggregation of information from neighboring nodes [4, 5, 2]. In an MPNN, node representations $\boldsymbol{h}^{v(l)} \in \mathbb{R}^d$ at the $l$-th layer are computed through $\mathsf{MP}^{(l)}$ as

$$\boldsymbol{h}^{v(l)} = \left(\mathsf{MP}^{(l)}\left(\boldsymbol{H}^{(l-1)}, \widetilde{\boldsymbol{A}}\right)\right)_v = \gamma^{(l)}\left(\boldsymbol{h}^{v(l-1)}, \underset{u \in \mathcal{N}(v)}{\mathrm{AGGR}}\left\{\phi^{(l)}\left(\boldsymbol{h}^{v(l-1)}, \boldsymbol{h}^{u(l-1)}, \tilde{a}^{uv}\right)\right\}\right) \tag{2}$$

where $\gamma^{(l)}$ and $\phi^{(l)}$ are differentiable *update* and *message* functions, respectively, $\mathrm{AGGR}\{\cdot\}$ is a permutation invariant *aggregation* function over the set of messages, and $\mathcal{N}(v)$ is the set of incoming neighbors of $v$. Borrowing this terminology, in the following we use the term *temporal message-passing* [40] for any function $\mathsf{TMP}^{(l)}$ that computes each $i$-th representation $\boldsymbol{h}_{t-i}^{(l)}$ from the sequence $\boldsymbol{h}_{t\text{-}T:t}^{(l-1)}$ by conditioning on the temporal dependencies defined by $\boldsymbol{T}$, i.e.,

$$\boldsymbol{h}_{t\text{-}T:t}^{(l)} = \mathsf{TMP}^{(l)}\left(\boldsymbol{h}_{t\text{-}T:t}^{(l-1)}, \boldsymbol{T}\right). \tag{3}$$

Examples of this function class include RNNs and TCNs. While the MP and TMP operators described previously are constrained to processing along a single dimension, the *spatiotemporal message-passing* layer $\mathsf{STMP}$ extends this capability to operate simultaneously across both spatial and temporal dimensions [41]. This allows the model to condition its output on both the graph and backward shift-operators $\widetilde{\boldsymbol{A}}$ and $\boldsymbol{T}$, respectively:

$$\boldsymbol{h}_{t\text{-}T:t}^{v(l)} = \left(\mathsf{STMP}^{(l)}\left(\boldsymbol{H}_{t\text{-}T:t}^{(l-1)}, \widetilde{\boldsymbol{A}}, \boldsymbol{T}\right)\right)_v. \tag{4}$$

**Over-squashing in GNNs** This term describes the compression of exponentially increasing information throughout the layers of a GNN [30], which particularly hinders long-range interactions [42]. A common way to assess over-squashing is by means of a sensitivity analysis through the Jacobian

$$\nabla^u \boldsymbol{h}^{v(L)} := \frac{\partial \boldsymbol{h}^{v(L)}}{\partial \boldsymbol{h}^{u(0)}} \in \mathbb{R}^{d \times d}, \tag{5}$$

whose spectral norm $\left\|\nabla^u \boldsymbol{h}^{v(L)}\right\|$ acts as a proxy to measure how much initial information at node $u$ can influence the representation computed at node $v$ after $L$ GNN processing layers [31, 32]. For a broad class of MPNNs, Di Giovanni et al. [43] obtained the following bound, which isolates the two contributions of architecture and graph structure to over-squashing.

**Theorem C.1** (from Di Giovanni et al. [43]). *Consider an MPNN with $L$ layers, with $c_\xi$ being the Lipschitz constant of the update function after activation $\xi$, and $\theta_m$ and $\theta_u$ being the maximal norms over all weight matrices in the message and update functions, respectively. For $v, u \in V$ we have:*

$$\left\| \nabla^u \boldsymbol{h}^{v(L)} \right\| \leq \underbrace{(c_\xi \theta_m)^L}_{\text{model}} \underbrace{\left( \mathbf{S}^L \right)_{uv}}_{\text{topology}},$$

*where* $\mathbf{S} := \frac{\theta_u}{\theta_m} \boldsymbol{I} + c_1 \operatorname{diag}\left( \widetilde{\boldsymbol{A}}^\top \mathbf{1} \right) + c_2 \widetilde{\boldsymbol{A}} \in \mathbb{R}^{N \times N}$, *is the message-passing matrix such that the Jacobian of the message function $\phi^{(l)}$ w.r.t. the target ($v$) and neighbor ($u$) node features has bounded norms $c_1$ and $c_2$, respectively.*

## 3  Information propagation and over-squashing in STGNNs

In line with previous works [41, 20], we consider STGNNs obtained by stacking $L$ STMP layers (Eq. 4), preceded and followed by differentiable encoding and decoding (readout) functions:

$$\boldsymbol{h}_t^{v(0)} = \mathsf{ENCODER}\left(\boldsymbol{x}_t^v\right), \qquad (6) \qquad\qquad \hat{\boldsymbol{y}}_t^v = \mathsf{READOUT}\left(\boldsymbol{h}_t^{v(L)}\right). \qquad (7)$$

Note that the encoder is applied independently to each node and time step, while the readout produces estimates for the label using only node representations associated with the last time step. Most existing STGNNs can be represented following this framework and differ primarily in the processing carried out in STMP. In the following, we consider STMP operators resulting from the composition of message-passing, i.e., MP, and sequence-processing, i.e., TMP, operators.

### 3.1  Spatiotemporal message-passing designs

A straightforward yet effective strategy to design an STMP layer is to factorize processing along the two dimensions with a sequential application of MP and TMP. This enables using existing operators, making the resulting STGNN easy to implement. We can write the $l$-th STMP layer as:

$$\boldsymbol{z}_{t-T:t}^{v(l)} = \mathsf{TMP}_{\times L_\mathsf{T}}^{(l)}\left(\boldsymbol{h}_{t-T:t}^{v(l-1)}, \boldsymbol{T}\right) \;\; \forall v \in \mathcal{V} \;\; (8) \quad \boldsymbol{h}_{t-j}^{v(l)} = \left(\mathsf{MP}_{\times L_\mathsf{S}}^{(l)}\left(\boldsymbol{Z}_{t-j}^{(l)}, \widetilde{\boldsymbol{A}}\right)\right)_v \;\; \forall j \in [0, T) \;\; (9)$$

where $\boldsymbol{z}_{t-T:t}^{v(l)} \in \mathbb{R}^{T \times d}$ is the sequence of intermediate representations resulting from node-level temporal encoding, and MP is applied independently (with shared parameters) across time steps. The subscript $\cdot_{\times L_\mathsf{T}}$ ($\cdot_{\times L_\mathsf{S}}$) concisely denotes a stack of $L_\mathsf{T}$ ($L_\mathsf{S}$) functions of the same family – with distinct parameters – each receiving as input representation the output from the preceding function in the stack. Although processing is decoupled within a single STMP layer, the resulting representations effectively incorporate information from the history of neighboring nodes.

Unlike the graph domain, where MPNNs have established themselves as the standard framework for processing relational data, the temporal domain lacks a unified approach that encompasses all architectures. Due to their architectural similarity to GNNs, in this work, we focus on TCNs, which allows for a more natural extension to spatiotemporal modeling and facilitates drawing analogies between temporal and graph-based representation learning.

**MPTCNs**  We call message-passing temporal convolutional networks (MPTCNs) those STGNNs obtained by combining TCNs and MPNNs following the framework defined in Eq. 8–9 [44, 21]. Specifically, the TMP operator is implemented as a causal convolution of a nonlinear filter with $P$ elements over the temporal dimension [45, 36, 16]. Causal convolutions can be expressed through a Toeplitz matrix formulation, which enables the analysis of their propagation dynamics. To formalize this, we introduce the lower-triangular, Toeplitz matrix $\mathbf{R} \in \{0,1\}^{T \times T}$ where $(\mathbf{R})_{ij} = 1$ if the input at time step $t - i$ influences the output at time step $t - j$. This matrix encodes temporal dependencies analogously to how the graph shift operator $\widetilde{\boldsymbol{A}}$ encodes spatial relationships in MPNNs. We define the $l$-th layer of a TCN $\mathsf{TC}^{(l)}$ as:

$$\boldsymbol{h}_{t-T:t}^{(l)} = \mathsf{TC}^{(l)}\left(\boldsymbol{h}_{t-T:t}^{(l-1)}, \mathbf{R}\right) = \sigma\left(\sum_{p=0}^{T} \operatorname{diag}_p(\mathbf{R})^\top \boldsymbol{h}_{t-T:t}^{(l-1)} \boldsymbol{W}_p^{(l)}\right) \qquad (10)$$

where each $\boldsymbol{W}_p^{(l)} \in \mathbb{R}^{d \times d}$ is a matrix of learnable weights, $\sigma$ is an element-wise activation function (e.g., ReLU), and $\operatorname{diag}_p(\mathbf{R})$ is the matrix obtained by zeroing all entries of $\mathbf{R}$ except those on its

$p$-th lower diagonal. In standard convolutional implementations, we employ $\mathbf{R} = \sum_{p=0}^{P-1} \boldsymbol{T}^p$, with $(\mathbf{R})_{ij} = 1$ only if $0 \leq i - j < P$, such that $\mathrm{diag}_p(\mathbf{R}) = \boldsymbol{T}^p$. This formulation reveals the structural parallels between temporal and spatial message passing, both conditioned on specialized operators ($\mathbf{R}$ and $\widetilde{\boldsymbol{A}}$, respectively) that encode the underlying topology.

***Time-and-space* vs *time-then-space*** By adjusting the number of outer layers $L$ and inner layers $L_\mathsf{T}$ and $L_\mathsf{S}$, we can control the degree of temporal-spatial integration while maintaining fixed total processing depth, $LL_\mathsf{T}$ and $LL_\mathsf{S}$, which we refer to as the *temporal* and *spatial budget* respectively. Fixing both budgets enables a fair experimental comparison between TTS and T&S variants. When $L = 1$, processing becomes fully decoupled, yielding the computationally efficient TTS approach that has gained recent prominence [35, 46, 47]. This efficiency arises because encoder-decoder architectures only require representations at time $t$ and layer $L$ for readout (see Eq. 7), allowing message passing on a single (static) graph with features $\boldsymbol{H}_t^{(L)}$. Thus, with equivalent parameter counts and layer depths, the TTS approach offers substantially reduced computational complexity for spatial processing, scaling with $\mathcal{O}(T)$ – a detailed discussion of the computational complexities is provided in Appendix C. This advantage is particularly valuable in practical applications, where temporal processing can occur asynchronously across nodes before being enriched with spatial context, enabling efficient distributed implementations [48]. However, T&S architectures may be more suitable and straightforward to adopt when the graph topology varies over time, with time steps associated with potentially different adjacency matrices.

## 3.2 Spatiotemporal over-squashing

While sensitivity analysis has become a standard approach to studying over-squashing in static GNNs [31, 32, 43, 49], it has been limited to graphs with static node features. We extend this analysis to STGNNs by examining the sensitivity of node representations after both temporal and spatial processing. In particular, we are interested in studying how information propagation across space and time affects the sensitivity of learned representations to initial node features at previous time steps. Considering that $\partial \boldsymbol{h}_t^{v(0)}/\partial \boldsymbol{x}_{t-i}^u = \boldsymbol{0}$ and $\partial \hat{\boldsymbol{y}}_t^v/\partial \boldsymbol{h}_{t-i}^{u(L)} = \boldsymbol{0}$ for each $i \neq 0, u \neq v$, we analyze the Jacobian between node features after a stack of $L$ STMP layers, i.e.,

$$\nabla_i^u \boldsymbol{h}_t^{v(L)} := \frac{\partial \boldsymbol{h}_t^{v(L)}}{\partial \boldsymbol{h}_{t-i}^{u(0)}} \in \mathbb{R}^{d \times d}. \tag{11}$$

This quantity differs conceptually from the simpler static-graph setting (Eq. 5), as the temporal dimension represents an additional propagation axis. Notably, in decoupled STMP functions (Eq. 8–9), information flows strictly along separate dimensions: TMP operates exclusively within the temporal domain, while MP processes only spatial relationships, preventing cross-dimensional interactions. Hence, given an STGNN of $L$ layers, for each layer $l \in [1, L]$, nodes $u, v \in \mathcal{V}$ and $i, j \in [0, T)$ we have

$$\frac{\partial \boldsymbol{h}_{t-j}^{v(l)}}{\partial \boldsymbol{h}_{t-i}^{u(l-1)}} = \underbrace{\frac{\partial \boldsymbol{h}_{t-j}^{v(l)}}{\partial \boldsymbol{z}_{t-j}^{u(l)}}}_{\text{space}} \underbrace{\frac{\partial \boldsymbol{z}_{t-j}^{u(l)}}{\partial \boldsymbol{h}_{t-i}^{u(l-1)}}}_{\text{time}} . \tag{12}$$

This factorization allows us to independently study the effects of temporal and spatial processing within each layer on the output representations. Moreover, in the TTS case where $L = 1$, this result provides us a factorized tool to investigate $\nabla_i^u \boldsymbol{h}_t^{v(L)}$ and measure how initial representations affect the final output. While the spatial component has been extensively studied in the literature on GNNs (as discussed in Sec. 6), the temporal dimension, especially the phenomenon of *temporal over-squashing*, remains less explored. In the following section, we investigate how temporal processing affects representation learning in TCNs, and consequently, MPTCNs. Following an incremental approach, we first discuss its effects on propagation dynamics in TCNs in the next section, and then extend the analysis to the spatiotemporal setting in Sec. 5.

## 4 Over-squashing in TCNs

To isolate the temporal dynamics, we focus exclusively on the temporal processing component by analyzing encoder-decoder networks constructed from $L_\mathsf{T}$ successive TC layers positioned between

encoding and readout operations. In these architectures, each additional layer expands the network's temporal receptive field, enabling the model to capture progressively longer-range dependencies in the input sequence. This hierarchical processing creates an information flow pattern that can be precisely characterized through the powers of the temporal topology matrix $\mathbf{R}$. Specifically, $\mathbf{R}^l$ represents the temporal receptive field at the $l$-th layer, where entry $(\mathbf{R}^l)_{ij}$ quantifies the number of distinct paths through which information can propagate from the input at time step $t - i$ to the representation at time step $t - j$ after $l$ layers of processing.

Understanding how information flows through these paths is crucial for analyzing the model's capacity to effectively leverage temporal context. To formalize this analysis, we investigate how the output representations are influenced by perturbations in the input at preceding time steps. The following theorem establishes a bound on this sensitivity, revealing how the temporal topology governs information flow across layers and drawing important parallels to the graph domain.

**Theorem 4.1.** *Consider a TCN with $L_\mathsf{T}$ successive* $\mathsf{TC}$ *layers as in Eq. 10, all with kernel size $P$, and assume that $\left\| \boldsymbol{W}_p^{(l)} \right\| \leq \mathsf{w}$ for all $p < P$ and $l \leq L_\mathsf{T}$, and that $|\sigma'| \leq c_\sigma$. For each $i, j \in [0, T)$, we have:*

$$\left\| \frac{\partial \boldsymbol{h}_{t-j}^{(L_\mathsf{T})}}{\partial \boldsymbol{h}_{t-i}^{(0)}} \right\| \leq \underbrace{(c_\sigma \mathsf{w})^{L_\mathsf{T}}}_{\text{model}} \underbrace{\left( \mathbf{R}^{L_\mathsf{T}} \right)_{ij}}_{\text{temporal topology}}.$$

Proof provided in Appendix A.1. Similar to the spatial case in Theorem C.1, this bound comprises two components: one dependent on model parameters and another on the temporal topology encoded in $\mathbf{R}$. Unlike spatial topologies, however, the temporal structure follows a specific, well-defined pattern that enables deeper theoretical analysis. The lower-triangular Toeplitz structure of $\mathbf{R}$ ensures that its powers maintain this structure [50]. This property leads to a distinctive pattern of influence distribution, formalized in the following proposition:

**Proposition 4.2.** *Let $\mathbf{R} \in \mathbb{R}^{T \times T}$ be a real, lower-triangular, Toeplitz band matrix with lower bandwidth $P - 1$, i.e., with $(\mathbf{R})_{ij} = r_{i-j}$ for $0 \leq i - j < P$, and $P \geq 2$, $r_1 \neq 0$, and $r_0 \neq 0$. Then for any $i > j$ we have $\left| \frac{(\mathbf{R}^l)_{j0}}{(\mathbf{R}^l)_{i0}} \right| \to 0$ as $l \to \infty$. In fact $\left| \frac{(\mathbf{R}^l)_{j0}}{(\mathbf{R}^l)_{i0}} \right| = \mathcal{O}(l^{-(i-j)})$. Informally, this means that the final token receives considerably more influence from tokens positioned earlier.*

Proof provided in Appendix A.2. Proposition 4.2 reveals a critical insight: **causal convolutions progressively diminish sensitivity to recent information while amplifying the influence of temporally distant inputs**. This creates a form of temporal over-squashing that inverts the pattern observed in MPNNs, where distant nodes suffer from reduced influence [30]. We show this graphically in Fig. 2a, which also displays the powers of the temporal topology matrix (a.1–a.2); a yellow color in matrix entry $(i, j)$ indicates strong influence exerted from time step $t-i$ to $t-j$. As more convolutional layers are stacked, the influence of temporally recent inputs diminishes relative to more distant ones. This behavior stems from the structure of causal convolutions, which incrementally incorporate more information into a fixed-length context vector. Indeed, causal convolutions propagate information along powers of a directed path graph. Over multiple layers, earlier time steps accumulate influence through an increasing number of propagation paths, while more recent inputs have fewer paths for propagating their initial information. Crucially, because causal convolutions are forward-only, each time step can preserve its information in the associated context vector through self-loops only, with a major impact on the last time step in the sequence.

When a model's receptive field exceeds the sequence length, i.e., $(P - 1)L_\mathsf{T} \gg T$, the earliest time step exerts disproportionate influence on the final output compared to any intermediate time step, mirroring the recently investigated *attention sink* effect in Transformers [51, 52]. This behavior directly undermines the locality bias that causal convolutions are designed to enforce, particularly in time series applications, where recent observations typically carry greater relevance. To achieve a more balanced receptive field that preserves local information while incorporating broader context, we can act by modifying $\mathbf{R}$, effectively implementing *temporal graph rewiring* analogous to techniques used to address over-squashing in MPNNs [31, 32].

**Temporal graph rewiring** Proposition 4.2 outlines that the influence from recent time steps progressively vanishes when the temporal topology (1) remains fixed across layers and (2) maintains a lower-triangular Toeplitz structure. We propose two rewiring approaches targeting these assumptions

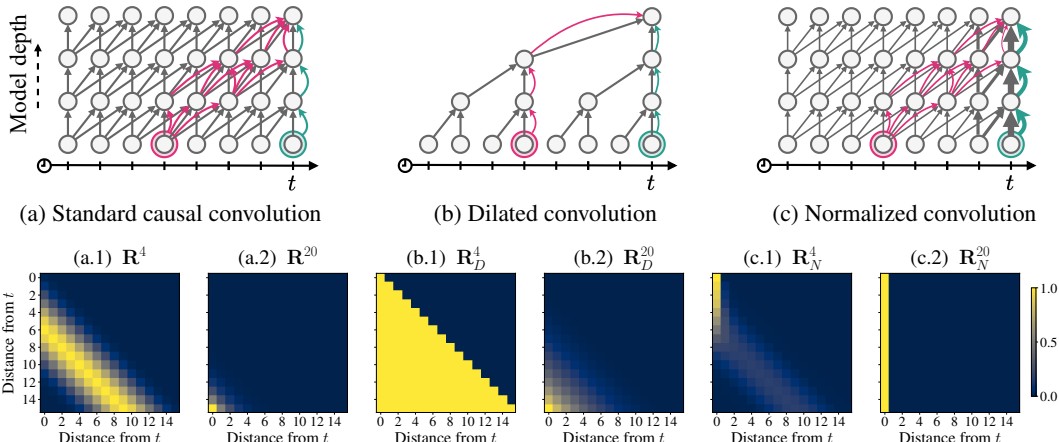

Figure 2: **Top row:** paths for information flow from the most recent and an earlier time step to the last-layer representation at time $t$. **Bottom row:** evolution of the temporal receptive field after 4 and 20 layers, seen through the powers of the temporal topology matrix. For standard ($\mathbf{R}$) and dilated ($\mathbf{R}_D$) convolution, the highest-influence region shifts towards the initial time step, while for row-normalized ($\mathbf{R}_N$) convolution, we observe a progressive shift to a uniform distribution across all time steps (first column). Entries are scaled matrix-wise in the range $[0, 1]$ for comparison purposes.

separately. Our first approach addresses the fixed topology assumption by employing different matrices $\mathbf{R}^{(l)}$ at each layer (each maintaining a lower-triangular Toeplitz structure). This modifies the Jacobian bound in Theorem 4.1 to depend on $\prod_{l=1}^{L_\mathsf{T}} \mathbf{R}^{(l)}$ rather than $\mathbf{R}^{L_\mathsf{T}}$. Dilated convolutions naturally implement this approach, applying filters with progressively increasing gaps $d^{(l)}$ (dilation rates) between elements [16] (Fig. 2b). These convolutions produce matrices $\mathbf{R}_D^{(l)}$ with nonzero entries $\left(\mathbf{R}_D^{(l)}\right)_{ij} = r_{i-j}$ only when $i - j = kd^{(l)}$ for $k \in [0, P)$. When $d^{(l)} = P^{l-1}$, the receptive field expands exponentially while ensuring $\left(\prod_l \mathbf{R}_D^{(l)}\right)_{p0} = 1$ for each $p \leq P^l$, hence distributing influence equally across all time steps in the receptive field (Fig. 2b.1). Thus, besides efficiency, dilated convolutions have the advantage of preserving local information better than standard convolutions. However, this strategy results in the trivial identity $\mathbf{R}_D^{(l)} = \boldsymbol{I}_T$ for $l > \log_P T$, with deeper architectures required to reset the dilation rate every $m$ layers, i.e., $d^{(l)} = P^{(l-1) \bmod m}$. While effective in practice, these resets reintroduce over-squashing patterns, as shown in Fig. 2b.2.

Our second approach targets the Toeplitz assumption by row-normalizing $\mathbf{R}$ to create $\mathbf{R}_N = \text{diag}(\mathbf{R}\mathbf{1})^{-1}\mathbf{R}$, where each entry $(\mathbf{R}_N)_{ij}$ is normalized by the number of edges from time step $t - i$ (Fig. 2c). This normalization maintains stronger influence from recent time steps while expanding the receptive field, with $\left(\mathbf{R}_N^l\right)_{i0}$ converging to 1 for all $i \in [0, T)$ as $l \to \infty$ (illustrated in Fig. 2c.1–c.2 and proven in Proposition A.2 in the appendix). Despite violating the Toeplitz structure, this approach remains computationally efficient by simply dividing the input at time step $t - i$ by $\min(i + 1, P)$. Nonetheless, this normalization primarily benefits the final time step prediction, making it particularly suitable for forecasting tasks where only the last output is used. For tasks requiring readout at intermediate time steps (e.g., imputation), the benefits of such a mitigation may be limited.

**Empirical validation** We empirically validate the effects of our proposed temporal convolution modifications through two synthetic sequence memory tasks: COPYFIRST and COPYLAST, where the goal is to output, respectively, the first or last observed value in a sequence of $T = 16$ random values sampled uniformly in $[0, 1]$. Since always predicting $0.5$ yields an MSE of $\approx 0.083$, we consider a task solved when the test MSE is lower than $0.001$ and report the success rate across multiple runs. We compare three TCNs architectures obtained by stacking $L_\mathsf{T}$ TC layers with different temporal convolution topologies: (1) $\mathbf{R}$, the standard causal convolution; (2) $\mathbf{R}_N$, where $\mathbf{R}$ is row-normalized; (3) $\mathbf{R}_D$, implementing dilated convolutions with dilation rates $d^{(l)} = P^{(l-1) \bmod m}$ and with $M = 4$. We set $P = 4$ in $\mathbf{R}$ and $\mathbf{R}_N$, and $P = 2$ in $\mathbf{R}_D$ and vary $L_\mathsf{T}$ from 1 to 20; Fig. 3 shows the simulation results. For small $L_\mathsf{T}$, the COPYFIRST task remains unsolved as the initial time step falls outside the receptive field, while standard convolutions fail on COPYLAST when $L_\mathsf{T} > 5$ due to

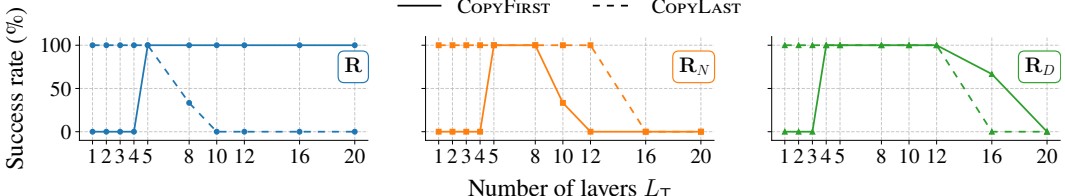

Figure 3: Success rate (%) on the tasks of copying the first or last observed value across different temporal topologies and number of layers $L_{\mathsf{T}}$.

the sink phenomenon. As network depth increases, performance degrades across all approaches, despite COPYFIRST remaining more tractable given the sink-induced bias toward earlier time steps, suggesting a fundamental connection to vanishing gradients [33].

## 5  Over-squashing in MPTCNs

Having analyzed over-squashing in temporal and spatial domains separately, we now integrate these insights to investigate information flow in MPTCNs, in which temporal and spatial processing are interleaved across layers. For our analysis, we consider the same class of message-passing functions employed by Di Giovanni et al. [43] in Theorem C.1, which generalizes many popular MPNNs [25, 53–55]. Following the space-time factorization established in Eq. 12, we observe that within a single layer, the sensitivity is simply the product of the spatial bound from Theorem C.1 and the temporal bound from Theorem 4.1. For TTS architectures with $L = 1$, this yields:

$$\left\| \nabla_i^u \boldsymbol{h}_t^{v(L)} \right\| \leq \underbrace{(c_\xi \theta_{\mathsf{m}})^{L_{\mathsf{S}}} \left( \mathbf{S}^{L_{\mathsf{S}}} \right)_{uv}}_{\text{space}} \underbrace{(c_\sigma \mathsf{w})^{L_{\mathsf{T}}} \left( \mathbf{R}^{L_{\mathsf{T}}} \right)_{i0}}_{\text{time}}, \tag{13}$$

This bound directly measures how input features influence the final representations used by the readout layer. However, this result only addresses the case where $L = 1$ (the TTS approach). The question remains: how does information propagate when temporal and spatial processing alternate multiple times ($L > 1$, the T&S approach)? To answer this question, we derive the following theorem that characterizes sensitivity across MPTCNs with any number of alternating processing blocks.

**Theorem 5.1.** *Consider an MPTCN with $L$ STMP layers, each consisting of $L_{\mathsf{T}}$ temporal (TMP) and $L_{\mathsf{S}}$ spatial (MP) layers as defined in Eq. 8–9. Assume that each TMP layer satisfies the conditions of Theorem 4.1, and each MP layer satisfies the assumptions in Theorem C.1. Then, for any $v, u \in \mathcal{V}$ and $i, j \in [0, T]$, the following holds:*

$$\left\| \frac{\partial \boldsymbol{h}_{t-j}^{v(L)}}{\partial \boldsymbol{h}_{t-i}^{u(0)}} \right\| \leq \underbrace{(c_\xi \theta_{\mathsf{m}})^{LL_{\mathsf{S}}} (c_\sigma \mathsf{w})^{LL_{\mathsf{T}}}}_{\text{model}} \underbrace{\left( \mathbf{S}^{LL_{\mathsf{S}}} \right)_{uv} \left( \mathbf{R}^{LL_{\mathsf{T}}} \right)_{ij}}_{\text{spatiotemporal topology}}.$$

The proof is provided in Appendix A.2. This result bounds the influence of input features $\boldsymbol{h}_{t-i}^{u(0)}$ on output representation $\boldsymbol{h}_{t-j}^{v(L)}$, revealing a clean separation between model parameters and topological factors, as well as between spatial and temporal components. Two key implications emerge from this factorization. First, the bound's multiplicative structure across space and time dimensions persists regardless of how many STMP layers are used. This means that redistributing the computational budget among outer layers $L$ and inner layers $L_{\mathsf{T}}$ and $L_{\mathsf{S}}$ does not alter the bound's characteristics. Therefore, **from the perspective of information propagation, TTS architectures ($L = 1$) are not inherently limited compared to T&S architectures ($L > 1$).** While this does not guarantee equivalence in expressivity or optimization dynamics, it provides a principled justification for adopting more computationally efficient TTS designs without compromising how information flows. Second, the theorem reveals that **spatiotemporal over-squashing in MPTCNs arises from the combined effects of spatial and temporal over-squashing**. This is evident in the spatiotemporal topology component, where both the spatial distance between nodes $u$ and $v$ and the temporal distance between time steps $t - i$ and $t - j$ contribute equally to potential bottlenecks. This insight carries significant practical implications: addressing over-squashing effectively requires targeting both dimensions

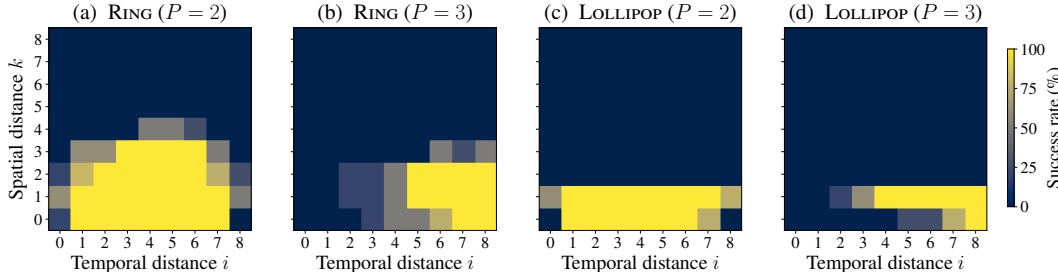

Figure 4: Success rate (%) of TTS MPTCNs on the ROCKETMAN dataset, where the goal is to copy the average value associated with $k$-hop neighbors at time step $t - i$. The tasks vary for the type of graph used (RING or LOLLIPOP) and size of $P$ (2 or 3).

simultaneously. Improving only one component – through either spatial or temporal graph rewiring alone – will prove insufficient if bottlenecks persist in the other dimension.

## 5.1 Empirical validation

To validate our theoretical results, we conducted experiments on both synthetic and real-world tasks that highlight the effects of spatiotemporal over-squashing. The reference MPTCN architecture used in the experiments features the Diffusion Convolution (DCNN) operator [54] as the MP layer.

**Synthetic environments** We design a synthetic memory task named ROCKETMAN where, given a graph and a time window of random values, the model must retrieve the average value at time step $t - i$ for nodes exactly $k$ hops away from a target node. We keep the input size constant across tasks and employ a TTS architecture with precisely enough layers in each dimension to span the entire input space, reporting success rates across multiple runs. The results, shown in Fig. 4, reveal two key patterns consistent with our theory. First, the task is significantly more challenging on the LOLLIPOP graph compared to the RING graph, confirming the known spatial over-squashing characteristics of these topologies [43]. Second, as the convolutional filter size $P$ increases, we observe that the task becomes more challenging for lower temporal distances, aligning perfectly with our analysis of TCNs. Complete experimental details, including comparative results for a T&S architecture, are provided in Appendix B, while in Appendix D we show results on TEMPORALNEIGHBOURSMATCH, an adaptation of the synthetic environment NEIGHBOURSMATCH [30] to the spatiotemporal setting.

**Real-world benchmarks** To verify that our theoretical insights extend to practical applications, we evaluated various MPTCNs of *fixed spatial and temporal budgets* $LL_S$ and $LL_T$ respectively, on three spatiotemporal forecasting benchmarks: METR-LA [19], PEMS-BAY [19], and EngRAD [40]. Tab. 1 presents the prediction errors in terms of mean absolute error (MAE). The results offer several insights that support our theoretical analysis. First, T&S and TTS approaches perform comparably on average, with the more efficient TTS models outperforming in the majority of cases. This empirical finding supports our theoretical conclusion that the TTS design is not inherently limited from an infor-

Table 1: Forecasting error (MAE) of MPTCNs with fixed budget in real-world benchmarks.

| | MODELS | $L$ | METR-LA | PEMS-BAY | EngRAD |
|---|---|---|---|---|---|
| MPTCN | **R** | 6 | $3.19_{\pm 0.02}$ | $1.66_{\pm 0.00}$ | $44.43_{\pm 0.41}$ |
| | | 3 | $3.19_{\pm 0.01}$ | $1.65_{\pm 0.01}$ | $43.83_{\pm 0.03}$ |
| | | 1 | $3.14_{\pm 0.02}$ | $1.63_{\pm 0.01}$ | $44.47_{\pm 0.42}$ |
| | $\mathbf{R}_N$ | 6 | $3.17_{\pm 0.02}$ | $1.65_{\pm 0.01}$ | $41.82_{\pm 0.38}$ |
| | | 3 | $3.17_{\pm 0.01}$ | $1.65_{\pm 0.00}$ | $41.78_{\pm 0.09}$ |
| | | 1 | $3.16_{\pm 0.01}$ | $1.65_{\pm 0.01}$ | $40.38_{\pm 0.08}$ |
| GWNet (orig.) | | | $3.02_{\pm 0.02}$ | $1.55_{\pm 0.01}$ | $40.50_{\pm 0.27}$ |
| GWNet TTS | | | $3.00_{\pm 0.01}$ | $1.57_{\pm 0.00}$ | $40.64_{\pm 0.29}$ |

mation propagation perspective, although additional factors beyond may also influence comparative performance. Second, in the EngRAD dataset, which requires larger filter sizes to cover the input sequence, the row-normalization approach consistently improves performance. This hints that temporal over-squashing affects standard convolutional models in practical scenarios where longer temporal contexts are needed, as our theory suggests. Finally, we compare Graph WaveNet (GWNet) [21] – a widely-used and more complex T&S architecture – against its TTS counterpart. Results show that our findings remain valid even when more sophisticated architectural components are involved.

**Combining spatial and temporal rewiring** In this experiment, we assess the combined effect of spatial and temporal graph rewiring in alleviating spatiotemporal over-squashing. We adopt FoSR [56] as the graph rewiring method and evaluate the forecasting error with and without row-normalized convolutions. We test the models on EngRAD for its symmetric topology, which makes rewiring more meaningful compared to traffic forecasting tasks, where spatial

Table 2: Forecasting error (MAE) of TTS MPTCNs with and without spatial and temporal graph rewiring in EngRAD.

|  | Original graph | FoSR rewiring | |
|---|---|---|---|
|  |  | w/ RGCN | w/ DCNN |
| $\mathbf{R}$ | $44.47_{\pm 0.42}$ | $43.78_{\pm 0.29}$ | $43.50_{\pm 0.08}$ |
| $\mathbf{R}_N$ | $40.38_{\pm 0.08}$ | $41.10_{\pm 0.11}$ | $40.30_{\pm 0.16}$ |

structure is rigidly defined by the underlying road network. Besides our original MPTCN implementation relying on DCNN as MP layer, we further consider RGCN [57], to weight differently the contribution of rewired edges. We report results in the TTS setting in Tab. 2. We can observe that combining both spatial and temporal rewiring yields the best performance in the original implementation using DCNN. In particular, rewiring in each dimension individually improves accuracy, with the temporal one contributing the largest marginal gain. This is consistent with our theoretical analysis, reinforcing that temporal bottlenecks are a significant limiting factor in STGNNs.

# 6 Related work

The issue of over-squashing in GNNs was first highlighted by Alon and Yahav [30], who showed that GNNs struggle to capture long-range dependencies in graphs with structural bottlenecks. Building on this, Topping et al. [31] introduced a sensitivity-based framework to identify over-squashing, which was later extended by Di Giovanni et al. [32, 43]. Two main strategies have emerged to alleviate this problem: graph re-wiring to enhance connectivity [58, 31, 56, 59], and architectural modifications to stabilize gradients [60–62, 49]. Notably, these efforts largely overlook the role of time-evolving node features. Similar challenges in modeling long-range dependencies have been studied in temporal architectures like RNNs, where vanishing and exploding gradients hinder effective learning [33, 34]. Solutions include enforcing orthogonality [63] or antisymmetry [64] in weight matrices, as well as designing specialized stable architectures [65]. Recently, the attention sink effect [51] has revealed a bias in Transformers towards early tokens [66], with Barbero et al. [52] linking this phenomenon to over-squashing in language models. In the spatiotemporal domain, Gao and Ribeiro [35] analyzed the expressive power of STGNNs, showing their capabilities and limits in distinguishing non-isomorphic graphs in both T&S and TTS settings, while Gravina et al. [67] propose a framework tailored to long-range tasks in continuous-time dynamic graphs. Yet, to our knowledge, no prior work has directly tackled the problem of spatiotemporal over-squashing in STGNNs, where the interaction between spatial and temporal dimensions introduces unique challenges for information propagation.

# 7 Conclusions

In this work, we have formally characterized spatiotemporal over-squashing in STGNNs, demonstrating its distinctions from the static case. Our analysis reveals that convolutional STGNNs counterintuitively favor information propagation from temporally distant points, offering key insights into their behavior. Despite their structural differences, we proved that both T&S and TTS paradigms equally suffer from this phenomenon, providing theoretical justification for computationally efficient implementations. Experiments on synthetic and real-world datasets confirm our theoretical framework's practical relevance. The insights from this study directly impact the design of effective spatiotemporal architectures. By bridging theory and practice, we contribute to a deeper understanding of STGNNs and provide principled guidance for their implementation.

**Limitations and future work**  We focus on factorized convolutional STGNNs, aligning with established GNN research while extending existing theoretical results to the spatiotemporal domain. This choice allows for clean theoretical bounds and a controlled setting applicable to both TTS and T&S variants. However, it also limits the generality of our results to models where cross-dimensional interactions are blocked. Extending our framework to models with joint space–time filters or recurrent STGNNs, which follow fundamentally different propagation dynamics, represents a valuable direction for future work. Finally, our sensitivity bounds are derived in the worst case and are therefore potentially conservative; deriving tighter, data-dependent estimates and conducting a systematic analysis of mitigation strategies remain promising directions for future research.

## Acknowledgments and Disclosure of Funding

The authors wish to thank Francesco Di Giovanni for the valuable feedback and collaboration during the initial phase of this research. M.B. and J.B. are partially supported by the EPSRC Turing AI World-Leading Research Fellowship No. EP/X040062/1 and EPSRC AI Hub No. EP/Y028872/1. C.A. is partly supported by the Swiss National Science Foundation under grant no. 204061 *HORD GNN: Higher-Order Relations and Dynamics in Graph Neural Networks* and the International Partnership Program of the Chinese Academy of Sciences under Grant 104GJHZ2022013GC.

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

# Appendix

## A Proofs

This appendix gathers the complete proofs of all theoretical results showcased in the main text. For clarity and ease of reference, we restate each proposition or theorem before providing its corresponding proof. We begin with the results pertaining to TCNs, followed by those related to the sensitivity analysis in MPTCNs.

### A.1 Sensitivity bound of TCNs

Theorem 4.1 establishes the sensitivity bound for a TCN composed by $L_\mathsf{T}$ stacked causal convolutional layers TC. The resulting inequality factorises into a model-dependent term $(c_\sigma \mathsf{w})^{L_\mathsf{T}}$ and a term $\mathbf{R}^{L_\mathsf{T}}$ dependent instead on the temporal topology. The proof proceeds by induction on the number of layers, bootstrapping from the single–layer Jacobian estimate we provide in Lemma A.1.

**Lemma A.1** (Single TC layer). *Consider a TC layer as in Eq. 10 with kernel size $P$, and assume that $\|\boldsymbol{W}_p\| \leq \mathsf{w}$ for all $p < P$, and that $|\sigma'| \leq c_\sigma$. For each $i, j \in [0, T)$, the following holds:*

$$\left\| \frac{\partial \boldsymbol{h}_{t-j}^{(1)}}{\partial \boldsymbol{h}_{t-i}^{(0)}} \right\| \leq c_\sigma \mathsf{w} \left( \mathbf{R} \right)_{ij}.$$

*Proof.* Let $\tilde{\boldsymbol{h}}_t^{(1)}$ be the pre-activation output of the TC layer, such that $\boldsymbol{h}_t^{(1)} = \sigma\big(\tilde{\boldsymbol{h}}_t^{(1)}\big)$. Since $\mathbf{R}$ is lower-triangular and Toeplitz and has lower bandwidth $P - 1$, i.e., $(\mathbf{R})_{ij} = r_{i-j}$ for $0 \leq i - j < P$, for indices $i, j \in [0, T)$, we have

$$\frac{\partial \tilde{\boldsymbol{h}}_{t-j}^{(1)}}{\partial \boldsymbol{h}_{t-i}^{(0)}} = \begin{cases} r_{i-j} \boldsymbol{W}_{i-j}, & 0 \leq i - j < P, \\ 0, & \text{otherwise.} \end{cases}$$

Applying the chain rule through the pointwise non-linearity $\sigma$ gives

$$\frac{\partial \boldsymbol{h}_{t-j}^{(1)}}{\partial \boldsymbol{h}_{t-i}^{(0)}} = \text{diag}\left( \sigma'\big( \tilde{\boldsymbol{h}}_{t-j}^{(1)} \big) \right) \frac{\partial \tilde{\boldsymbol{h}}_{t-j}^{(1)}}{\partial \boldsymbol{h}_{t-i}^{(0)}}.$$

Applying the sub-multiplicative property of the spectral norm together with the bounds $|\sigma'| \leq c_\sigma$ and $\|\boldsymbol{W}_p\| \leq \mathsf{w}$ yields

$$\begin{aligned} \left\| \frac{\partial \boldsymbol{h}_{t-j}^{(1)}}{\partial \boldsymbol{h}_{t-i}^{(0)}} \right\| &\leq \left\| \text{diag}\left( \sigma'\big( \tilde{\boldsymbol{h}}_{t-j}^{(1)} \big) \right) \right\| \left\| \frac{\partial \tilde{\boldsymbol{h}}_{t-j}^{(1)}}{\partial \boldsymbol{h}_{t-i}^{(0)}} \right\| \\ &\leq c_\sigma \mathsf{w} |r_{i-j}| \\ &= c_\sigma \mathsf{w} \left( \mathbf{R} \right)_{ij}, \end{aligned}$$

which proves the theorem. $\square$

We now use the result in Lemma A.1 to prove the bound of Theorem 4.1 for a TCN obtained by stacking $L_\mathsf{T}$ TC layers.

**Theorem 4.1.** *Consider a TCN with $L_\mathsf{T}$ successive TC layers as in Eq. 10, all with kernel size $P$, and assume that $\big\|\boldsymbol{W}_p^{(l)}\big\| \leq \mathsf{w}$ for all $p < P$ and $l \leq L_\mathsf{T}$, and that $|\sigma'| \leq c_\sigma$. For each $i, j \in [0, T)$, we have:*

$$\left\| \frac{\partial \boldsymbol{h}_{t-j}^{(L_\mathsf{T})}}{\partial \boldsymbol{h}_{t-i}^{(0)}} \right\| \leq \underbrace{(c_\sigma \mathsf{w})^{L_\mathsf{T}}}_{\text{model}} \underbrace{\big( \mathbf{R}^{L_\mathsf{T}} \big)_{ij}}_{\text{temporal topology}}.$$

*Proof.* We fix the number of stacked TC layers to be $l = L_\mathsf{T}$ and prove the bound by induction on $l$. For the base case $l = 1$, Lemma A.1 gives

$$\left\| \frac{\partial \boldsymbol{h}_{t-j}^{(1)}}{\partial \boldsymbol{h}_{t-i}^{(0)}} \right\| \leq c_\sigma \mathsf{w}(\mathbf{R})_{ij} = (c_\sigma \mathsf{w})^1 (\mathbf{R}^1)_{ij}, \tag{4.1.1}$$

so the claim holds for one layer. We assume the bound is true for $l - 1$:

$$\left\| \frac{\partial \boldsymbol{h}_{t-k}^{(l-1)}}{\partial \boldsymbol{h}_{t-i}^{(0)}} \right\| \leq (c_\sigma \mathsf{w})^{l-1} (\mathbf{R}^{l-1})_{ik} \qquad \forall\, i, k. \tag{4.1.2}$$

For $l$ layers, the chain rule gives

$$\left\| \frac{\partial \boldsymbol{h}_{t-j}^{(l)}}{\partial \boldsymbol{h}_{t-i}^{(0)}} \right\| = \left\| \sum_{k=0}^{T-1} \frac{\partial \boldsymbol{h}_{t-j}^{(l)}}{\partial \boldsymbol{h}_{t-k}^{(l-1)}} \frac{\partial \boldsymbol{h}_{t-k}^{(l-1)}}{\partial \boldsymbol{h}_{t-i}^{(0)}} \right\| \leq \sum_{k=0}^{T-1} \left\| \frac{\partial \boldsymbol{h}_{t-j}^{(l)}}{\partial \boldsymbol{h}_{t-k}^{(l-1)}} \right\| \left\| \frac{\partial \boldsymbol{h}_{t-k}^{(l-1)}}{\partial \boldsymbol{h}_{t-i}^{(0)}} \right\|,$$

where the first factor is the base case of a single layer (4.1.1), and the second one is the induction hypothesis (4.1.2). Substituting these bounds gives

$$\left\| \frac{\partial \boldsymbol{h}_{t-j}^{(l)}}{\partial \boldsymbol{h}_{t-i}^{(0)}} \right\| \leq \sum_{k=0}^{T-1} \left( (c_\sigma \mathsf{w})(\mathbf{R})_{kj} (c_\sigma \mathsf{w})^{l-1} (\mathbf{R}^{l-1})_{ik} \right)$$

$$= (c_\sigma \mathsf{w})^l \sum_{k=0}^{T-1} (\mathbf{R}^{l-1})_{ik} (\mathbf{R})_{kj}$$

$$= (c_\sigma \mathsf{w})^l (\mathbf{R}^l)_{ij},$$

which proves the induction. □

## A.2 Asymptotic of $\mathbf{R}$ and sensitivity in MPTCNs

In the following, we provide a detailed analysis of the asymptotic behavior of the temporal topology matrix, which characterizes information propagation across time steps in both TCNs and MPTCNs. Afterwards, we provide the sensitivity bound and related proof for the latter.

### A.2.1 Asymptotic behavior of $\mathbf{R}$ and $\mathbf{R}_N$

**Proposition 4.2.** *Let $\mathbf{R} \in \mathbb{R}^{T \times T}$ be a real, lower-triangular, Toeplitz band matrix with lower bandwidth $P - 1$, i.e., with $(\mathbf{R})_{ij} = r_{i-j}$ for $0 \leq i - j < P$, and $P \geq 2$, $r_1 \neq 0$, and $r_0 \neq 0$. Then for any $i > j$ we have $\left| \frac{(\mathbf{R}^l)_{j0}}{(\mathbf{R}^l)_{i0}} \right| \to 0$ as $l \to \infty$. In fact $\left| \frac{(\mathbf{R}^l)_{j0}}{(\mathbf{R}^l)_{i0}} \right| = \mathcal{O}(l^{-(i-j)})$. Informally, this means that the final token receives considerably more influence from tokens positioned earlier.*

*Proof.* Write $\mathbf{R} = r_0 \mathbf{I} + \mathbf{N}$, where $\mathbf{N} := \mathbf{R} - r_0 \mathbf{I}$ is strictly lower-triangular. Because $\mathbf{I}$ and $\mathbf{N}$ commute, the binomial expansion for commuting matrices gives

$$\mathbf{R}^l = r_0^l \mathbf{I} + \sum_{k=1}^{T-1} \binom{l}{k} r_0^{l-k} \mathbf{N}^k, \qquad l \in \mathbb{N},$$

the sum truncating at $T - 1$ since $\mathbf{N}^T = \mathbf{0}$. For a fixed row index $i \geq 1$, the strictly lower-triangular structure implies $(\mathbf{N}^k)_{i0} = 0$ whenever $k > i$, thus

$$(\mathbf{R}^l)_{i0} = \sum_{k=1}^{i} \binom{l}{k} r_0^{l-k} (\mathbf{N}^k)_{i0}.$$

With $k$ fixed, $\binom{l}{k} \sim l^k / k!$ as $l \to \infty$ since $\lim_{l \to \infty} \frac{\binom{l}{k}}{l^k} = \frac{1}{k!}$, so with some abuse of notation

$$(\mathbf{R}^l)_{i0} \sim \sum_{k=1}^{i} \frac{l^k}{k!} r_0^{l-k} (\mathbf{N}^k)_{i0} = r_0^l \sum_{k=1}^{i} \frac{l^k}{k! r_0^k} (\mathbf{N}^k)_{i0}, \qquad l \to \infty.$$

Note that it is the product of $r_0^l$ and a degree $i$ polynomial in $l$, therefore in order to see the behaviour as $l \to \infty$ it suffices to study the leading term $\frac{l^i}{i! r_0^i} (\mathbf{N}^i)_{i0}$ and the factor of $r_0^l$. Additionally, due to

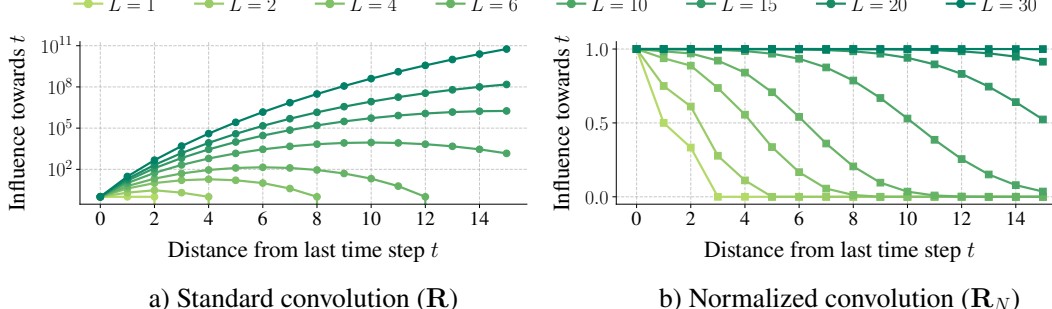

Figure 5: Sensitivity of last-layer representations associated with last time step $t$ to earlier ones in TCNs with $L$ layers and kernel size $P = 3$. The values correspond to entries $(\mathbf{R}^L)_{i0}$ for the standard convolution (a) and $(\mathbf{R}_N^L)_{i0}$ for the normalized convolution (b), with $i \geq 0$ being the backward distance from $t$. As depth increases, the standard convolution favors information from earlier steps, while the normalized version asymptotically approaches uniform sensitivity across all steps.

the strictly lower triangular structure of $\mathbf{N}$, we have $(\mathbf{N}^i)_{i0} = r_1^i$. This follows from the fact that, in a directed acyclic graph as the one described by $\mathbf{N}$, there exists a unique directed path of length $i$ from the $i$-th node to the sink node 0. Each edge along this path contributes a multiplicative factor of $r_1$, resulting in a total weight of $r_1^i$ for the path. Applying the same reasoning with $i$ replaced by $j < i$, we can study the ratio

$$\left| \frac{(\mathbf{R}^l)_{j0}}{(\mathbf{R}^l)_{i0}} \right| \sim \frac{\frac{l^j}{j!} r_0^{l-j} r_1^j}{\frac{l^i}{i!} r_0^{l-i} r_1^i} = \frac{1}{l^{i-j}} \frac{i! r_0^{i-j}}{j! r_1^{i-j}} = \mathcal{O}\left(\frac{1}{l^{i-j}}\right) \xrightarrow{l \to \infty} 0,$$

which proves the proposition. $\qquad\square$

The proposition that follows shows that the powers $\mathbf{R}_N^k$ converge to the rank–one matrix $\mathbf{1}e_0^\top$. In probabilistic terms, the normalization turns the temporal topology into an absorbing Markov chain whose unique absorbing state is the last time step. Consequently, the influence of any time step $t - i$ concentrates on $t$ as the depth grows, eventually reaching a uniform sensitivity of the last time step over the entire sequence. Fig. 5 shows a comparison between the sensitivity curves as a function of the (backward) temporal distance from the last time step for increasingly deeper TCNs in the case of standard (a) and normalized (b) convolutions.

**Proposition A.2.** *Let $\mathbf{R} \in \mathbb{R}_{\geq 0}^{T \times T}$ be a positive, real, lower–triangular Toeplitz band matrix with lower bandwidth $P - 1$, i.e., with $(\mathbf{R})_{ij} = r_{i-j}$ for $0 \leq i - j < P$, and let $P \geq 2$, $r_1 \neq 0$, and $r_0 \neq 0$. We define $\mathbf{R}_N := \mathrm{diag}(\mathbf{R}\mathbf{1})^{-1}\mathbf{R}$ to be the row–normalized matrix $\mathbf{R}$ ($\mathbf{1}$ is the all–ones vector). Then*

$$\lim_{k \to \infty} \mathbf{R}_N^k = \mathbf{1}e_0^\top,$$

*i.e. every row of $\mathbf{R}_N^k$ converges to $e_0^\top = (1, 0, \ldots, 0)$.*

*Proof.* Since $\mathbf{R}_N$ is a stochastic matrix (every row sum to 1 by construction), we can interpret $\mathbf{R}_N$ as the transition matrix of a Markov chain on the finite state space $\{0, 1, \ldots, T - 1\}$. Because $\mathbf{R}$ is lower-triangular, we have that state 0 is *absorbing*, i.e. $(\mathbf{R}_N)_{00} = r_0/r_0 = 1$, hence once the chain reaches 0 it never leaves. Moreover, since $r_1 \neq 0$, for every $i > 0$ and $j = i - 1$ there is a positive probability of moving one step closer to 0, i.e.,

$$(\mathbf{R}_N)_{ij} = \frac{r_1}{\sum_{p=0}^{\min\{P-1,i\}} r_p} > 0,$$

hence all states $1, 2, \ldots, T - 1$ are *transient*. For a finite absorbing Markov chain with a single absorbing state $\{0\}$, every state eventually reaches the absorbing state 0 with probability 1 (Theorem 11.3 from [68]). Thus,

$$\lim_{k \to \infty} \mathbf{R}_N^k = \mathbf{1}e_0^\top,$$

where the $i$-th row of the limit contains the absorption probabilities starting from state $i$ and are all equal to 1 for state 0 and to 0 for the others. $\qquad\square$

### A.2.2 Sensitivity bounds of MPTCNs

In this subsection, we aim to prove the sensitivity bounds of a MPTCN obtained by stacking $L$ STMP layers defined as in Eq. 8–9, where the temporal processing TMP takes the form of a temporal causal convolution, denoted by TC, as defined in Eq. 10.

In line with previous work by Di Giovanni et al. [43], for spatial processing, we consider as MP the following family of MPNNs:

$$\boldsymbol{h}^{v(l)} = \xi \left( \Theta_{\mathsf{U}}^{(l)} \boldsymbol{h}^{v(l-1)} + \Theta_{\mathsf{M}}^{(l)} \sum_{u \in \mathcal{N}(v)} \tilde{a}^{uv} \phi^{(l)} \left( \boldsymbol{h}^{v(l-1)}, \boldsymbol{h}^{u(l-1)} \right) \right) \tag{14}$$

where $\Theta_{\mathsf{U}}^{(l)}, \Theta_{\mathsf{M}}^{(l)} \in \mathbb{R}^{d \times d}$ are matrices of learnable weights, $\phi^{(l)}$ is a $\mathcal{C}^1$ function, and $\xi$ is a pointwise nonlinear activation function. This class includes common MPNNs, such as GCN [53], DCNN [54], GIN [25], and GatedGCN [55]. For this class of function, we make the following assumptions:

**Assumption A.3.** Given an MPNN with $L$ layers, each as in Eq. 14, we assume for each layer $l$ that $|\xi'| \le c_\xi$, $\left\| \Theta_{\mathsf{U}}^{(l)} \right\| \le \theta_{\mathsf{u}}$, and $\left\| \Theta_{\mathsf{M}}^{(l)} \right\| \le \theta_{\mathsf{m}}$. We further assume the Jacobian of the message function $\phi^{(l)}$ w.r.t. the target ($v$) and neighbor ($u$) node features to be bounded as $\left\| \partial \phi^{(l)} / \partial \boldsymbol{h}^{v(l-1)} \right\| \le c_1$ and $\left\| \partial \phi^{(l)} / \partial \boldsymbol{h}^{u(l-1)} \right\| \le c_2$.

Given an MPNN as defined in Eq. 14 and for which Assumption A.3 holds, Di Giovanni et al. [43] established the following sensitivity bound.

**Theorem C.1** (from Di Giovanni et al. [43]). *Consider an MPNN with $L$ layers, with $c_\xi$ being the Lipschitz constant of the update function after activation $\xi$, and $\theta_{\mathsf{m}}$ and $\theta_{\mathsf{u}}$ being the maximal norms over all weight matrices in the message and update functions, respectively. For $v, u \in V$ we have:*

$$\left\| \nabla^u \boldsymbol{h}^{v(L)} \right\| \le \underbrace{(c_\xi \theta_{\mathsf{m}})^L}_{\text{model}} \underbrace{\left( \mathbf{S}^L \right)_{uv}}_{\text{topology}},$$

*where $\mathbf{S} := \frac{\theta_{\mathsf{u}}}{\theta_{\mathsf{m}}} \boldsymbol{I} + c_1 \operatorname{diag} \left( \widetilde{\boldsymbol{A}}^\top \mathbf{1} \right) + c_2 \widetilde{\boldsymbol{A}} \in \mathbb{R}^{N \times N}$, is the message-passing matrix such that the Jacobian of the message function $\phi^{(l)}$ w.r.t. the target ($v$) and neighbor ($u$) node features has bounded norms $c_1$ and $c_2$, respectively.*

As done in the previous section to prove the bound for TCNs, we proceed by induction on the number of layers $L$, and start the analysis by establishing the bound for a single-layer MPTCN in Lemma A.4. The proof takes advantage of the result for TCNs we demonstrated in the previous section (Theorem 4.1) and Theorem C.1 by Di Giovanni et al. [43] for MPNNs.

**Lemma A.4** (Sensitivity bound TTS MPTCN). *Consider a TTS MPTCN ($L = 1$) with $L_{\mathsf{T}}$ temporal (TMP) layers and $L_{\mathsf{S}}$ spatial (MP) as defined in Eq. 8–9. Assume that each TMP layer satisfies the conditions of Theorem 4.1, and each MP layer satisfies the assumptions in Theorem C.1. Then, for any $v, u \in \mathcal{V}$ and $i, j \in [0, T)$, the following holds:*

$$\left\| \frac{\partial \boldsymbol{h}_{t-j}^{v(1)}}{\partial \boldsymbol{h}_{t-i}^{u(0)}} \right\| \le \underbrace{(c_\xi \theta_{\mathsf{m}})^{L_{\mathsf{S}}} \left( \mathbf{S}^{L_{\mathsf{S}}} \right)_{uv}}_{\text{space}} \underbrace{(c_\sigma \mathsf{w})^{L_{\mathsf{T}}} \left( \mathbf{R}^{L_{\mathsf{T}}} \right)_{ij}}_{\text{time}}.$$

*Proof.* Fix $u, v \in \mathcal{V}$ and $i, j \in [0, T)$. One STMP layer first applies the $L_{\mathsf{T}}$ TMP layers node–wise, and then the $L_{\mathsf{S}}$ MP layers time-wise. Thus, for the chain rule, we have

$$\frac{\partial \boldsymbol{h}_{t-j}^{v(1)}}{\partial \boldsymbol{h}_{t-i}^{u(0)}} = \sum_{w \in \mathcal{V}} \sum_{k=0}^{T-1} \frac{\partial \boldsymbol{h}_{t-j}^{v(1)}}{\partial \boldsymbol{z}_{t-k}^{w(L_{\mathsf{T}})}} \frac{\partial \boldsymbol{z}_{t-k}^{w(L_{\mathsf{T}})}}{\partial \boldsymbol{h}_{t-i}^{u(0)}}. \tag{A.4.1}$$

Every MP layer processes each time step separately, hence $\partial \boldsymbol{h}_{t-j}^{v(1)} / \partial \boldsymbol{z}_{t-k}^{w(L_{\mathsf{T}})} = 0$ unless $k = j$. Similarly, every TMP layer processes each node separately, hence $\partial \boldsymbol{z}_{t-j}^{w(L_{\mathsf{T}})} / \partial \boldsymbol{h}_{t-i}^{u(0)} = 0$ unless $w = u$. Both sums in Eq. A.4.1 therefore collapse, giving

$$\frac{\partial \boldsymbol{h}_{t-j}^{v(1)}}{\partial \boldsymbol{h}_{t-i}^{u(0)}} = \frac{\partial \boldsymbol{h}_{t-j}^{v(1)}}{\partial \boldsymbol{z}_{t-j}^{u(L_{\mathsf{T}})}} \frac{\partial \boldsymbol{z}_{t-j}^{u(L_{\mathsf{T}})}}{\partial \boldsymbol{h}_{t-i}^{u(0)}}.$$

Using sub-multiplicativity of the spectral norm, and considering the bounds from Theorem C.1 for MP layers and Theorem 4.1 for TC layers, we have

$$\left\|\frac{\partial \boldsymbol{h}_{t-j}^{v(1)}}{\partial \boldsymbol{h}_{t-i}^{u(0)}}\right\| \le \left\|\frac{\partial \boldsymbol{h}_{t-j}^{v(1)}}{\partial \boldsymbol{z}_{t-j}^{u(L_\mathsf{T})}}\right\| \left\|\frac{\partial \boldsymbol{z}_{t-j}^{u(L_\mathsf{T})}}{\partial \boldsymbol{h}_{t-i}^{u(0)}}\right\|$$

$$\le (c_\xi \theta_\mathsf{m})^{L_\mathsf{S}} \left(\mathbf{S}^{L_\mathsf{S}}\right)_{uv} (c_\sigma \mathsf{w})^{L_\mathsf{T}} \left(\mathbf{R}^{L_\mathsf{T}}\right)_{ij}. \qquad \square$$

Building on this result for a single-layer MPTCN, we extend the sensitivity bound for a MPTCN of $L$ layers in the following theorem.

**Theorem 5.1.** *Consider an MPTCN with $L$ STMP layers, each consisting of $L_\mathsf{T}$ temporal (TMP) and $L_\mathsf{S}$ spatial (MP) layers as defined in Eq. 8–9. Assume that each TMP layer satisfies the conditions of Theorem 4.1, and each MP layer satisfies the assumptions in Theorem C.1. Then, for any $v, u \in \mathcal{V}$ and $i, j \in [0, T)$, the following holds:*

$$\left\|\frac{\partial \boldsymbol{h}_{t-j}^{v(L)}}{\partial \boldsymbol{h}_{t-i}^{u(0)}}\right\| \le \underbrace{(c_\xi \theta_\mathsf{m})^{L L_\mathsf{S}} (c_\sigma \mathsf{w})^{L L_\mathsf{T}}}_{\text{model}} \underbrace{\left(\mathbf{S}^{L L_\mathsf{S}}\right)_{uv} \left(\mathbf{R}^{L L_\mathsf{T}}\right)_{ij}}_{\text{spatiotemporal topology}}.$$

*Proof.* We fix the number of STMP layers to be $l = L$ and prove the bound by induction on $l$. The base case where $l = 1$ follows directly from Lemma A.4, i.e.,

$$\left\|\frac{\partial \boldsymbol{h}_{t-j}^{v(1)}}{\partial \boldsymbol{h}_{t-i}^{u(0)}}\right\| \le (c_\xi \theta_\mathsf{m})^{L_\mathsf{S}} \left(\mathbf{S}^{L_\mathsf{S}}\right)_{uv} (c_\sigma \mathsf{w})^{L_\mathsf{T}} \left(\mathbf{R}^{L_\mathsf{T}}\right)_{ij} \qquad (5.1.1)$$

We assume the bound holds for $l - 1$ stacked STMP blocks, for all $u, w \in \mathcal{V}$ and $i, k \in [0, T)$:

$$\left\|\frac{\partial \boldsymbol{h}_{t-k}^{w(l-1)}}{\partial \boldsymbol{h}_{t-i}^{u(0)}}\right\| \le (c_\xi \theta_\mathsf{m})^{(l-1)L_\mathsf{S}} (c_\sigma \mathsf{w})^{(l-1)L_\mathsf{T}} \left(\mathbf{S}^{(l-1)L_\mathsf{S}}\right)_{uw} \left(\mathbf{R}^{(l-1)L_\mathsf{T}}\right)_{ik}. \qquad (5.1.2)$$

For the $l$-th layer, we apply the chain rule, the triangle inequality, and sub-multiplicativity:

$$\left\|\frac{\partial \boldsymbol{h}_{t-j}^{v(l)}}{\partial \boldsymbol{h}_{t-i}^{u(0)}}\right\| \le \sum_{w \in \mathcal{V}} \sum_{k=0}^{T-1} \left\|\frac{\partial \boldsymbol{h}_{t-j}^{v(l)}}{\partial \boldsymbol{h}_{t-k}^{w(l-1)}}\right\| \left\|\frac{\partial \boldsymbol{h}_{t-k}^{w(l-1)}}{\partial \boldsymbol{h}_{t-i}^{u(0)}}\right\|.$$

The first term is the base case of a single outer layer (5.1.1), and the right term is our induction hypothesis (5.1.2), which combined with the triangular inequality proves the induction:

$$\left\|\frac{\partial \boldsymbol{h}_{t-j}^{v(l)}}{\partial \boldsymbol{h}_{t-i}^{u(0)}}\right\| \le \sum_{w \in \mathcal{V}} \sum_{k=0}^{T-1} \left((c_\xi \theta_\mathsf{m})^{L_\mathsf{S}} (c_\sigma \mathsf{w})^{L_\mathsf{T}} \left(\mathbf{S}^{L_\mathsf{S}}\right)_{wv} \left(\mathbf{R}^{L_\mathsf{T}}\right)_{kj}\right.$$

$$\left.(c_\xi \theta_\mathsf{m})^{(l-1)L_\mathsf{S}} (c_\sigma \mathsf{w})^{(l-1)L_\mathsf{T}} \left(\mathbf{S}^{(l-1)L_\mathsf{S}}\right)_{uw} \left(\mathbf{R}^{(l-1)L_\mathsf{T}}\right)_{ik}\right)$$

$$= (c_\xi \theta_\mathsf{m})^{l L_\mathsf{S}} (c_\sigma \mathsf{w})^{l L_\mathsf{T}} \sum_{w \in \mathcal{V}} \sum_{k=0}^{T-1} \left(\mathbf{S}^{L_\mathsf{S}}\right)_{wv} \left(\mathbf{S}^{(l-1)L_\mathsf{S}}\right)_{uw} \left(\mathbf{R}^{L_\mathsf{T}}\right)_{kj} \left(\mathbf{R}^{(l-1)L_\mathsf{T}}\right)_{ik}$$

$$= (c_\xi \theta_\mathsf{m})^{l L_\mathsf{S}} (c_\sigma \mathsf{w})^{l L_\mathsf{T}} \left(\mathbf{S}^{(l-1)L_\mathsf{S}} \mathbf{S}^{L_\mathsf{S}}\right)_{uv} \left(\mathbf{R}^{(l-1)L_\mathsf{T}} \mathbf{R}^{L_\mathsf{T}}\right)_{ij}$$

$$= (c_\xi \theta_\mathsf{m})^{l L_\mathsf{S}} (c_\sigma \mathsf{w})^{l L_\mathsf{T}} \left(\mathbf{S}^{l L_\mathsf{S}}\right)_{uv} \left(\mathbf{R}^{l L_\mathsf{T}}\right)_{ij}.$$

Induction on $l$ establishes the inequality for every $l \le L$; in particular for $l = L$, which is exactly the claimed bound. $\qquad \square$

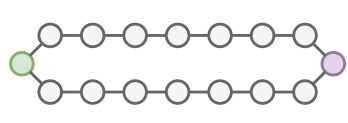
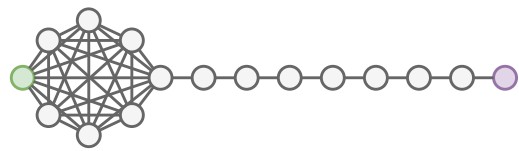

|         (a) RING          |        (b) LOLLIPOP        |

Figure 6: RING and LOLLIPOP graphs used in the synthetic experiments. We highlight in green the target node ○, and show an example of a source node ○ when the spatial distance $k$ is equal to the graph diameter.

# B Experiments

All numerical simulations are performed on regression tasks. For experiments on real-world data, the goal is $H$-steps-ahead forecasting. Given a window of $T$ past observations, the forecasting task consists of predicting the $H$ future observations at each node, i.e., $\boldsymbol{y}_t^v = \boldsymbol{x}_{t:t+H}^v$. We consider families of (parametric) models $f_{\boldsymbol{\theta}}$ such that

$$\hat{\boldsymbol{x}}_{t:t+H}^v = \left( f_{\boldsymbol{\theta}} \left( \boldsymbol{X}_{t\text{-}T:t}, \boldsymbol{A}, \boldsymbol{T} \right) \right)_v, \quad \forall\, v \in \mathcal{V}, \tag{15}$$

where $\boldsymbol{\theta}$ is the set of learnable parameters and $\hat{\boldsymbol{x}}_{t:t+H}^v$ are the forecasted values at node $v$ for the interval $[t, t+H)$. The parameters are optimized using an element-wise cost function, e.g., the MSE:

$$\hat{\boldsymbol{\theta}} = \arg\min_{\boldsymbol{\theta}} \frac{1}{NH} \sum_{v \in \mathcal{V}} \sum_{i=0}^{H-1} \left\| \hat{\boldsymbol{x}}_{t+i}^v - \boldsymbol{x}_{t+i}^v \right\|_2^2. \tag{16}$$

**Software & Hardware**   All the code used for the experiments has been developed with Python [69] and relies on the following open-source libraries: PyTorch [70]; PyTorch Geometric [71]; Torch Spatiotemporal [72]; PyTorch Lightning [73]; Hydra [74]; Numpy [75]. We relied on Weights & Biases [76] for tracking and logging experiments. The code to reproduce the experiments is available at github.com/marshka/spatiotemporal-oversquashing.

All experiments were conducted on a workstation running Ubuntu 22.04.5 LTS, equipped with two AMD EPYC 7513 CPUs and four NVIDIA RTX A5000 GPUs, each with 24 GB of memory. To accelerate the experimental process, multiple runs were executed in parallel, with shared access to both CPU and GPU resources. This setup introduces variability in execution times, even under identical experimental configurations. On average, experiments involving real-world datasets required approximately 1 to 2 hours per run, while synthetic experiments – when not terminated early due to the early-stopping criterion – completed within 20 minutes.

**Datasets**   We begin by introducing a set of synthetic datasets and tasks specifically designed to highlight the effects of over-squashing in both space and time.

**COPYFIRST**  Each sequence $\boldsymbol{x}_{t\text{-}T:t}$ consists of $T = 16$ time steps, with values sampled uniformly from the $[0, 1]$ interval. The task is to predict the first element in the sequence, i.e., $\boldsymbol{x}_{t\text{-}T+1}$. We generate 20,000 sequences for training, 320 for validation, and 500 for testing.

**COPYLAST**  This task is analogous to COPYFIRST, but the model is required to predict the last value in the sequence, i.e., $\boldsymbol{x}_t$. Together with COPYFIRST, these are the datasets used in the experiments in Sec.4.

Table 3: Statistics of the datasets and considered sliding-window parameters.

| Datasets | Type | Nodes | Edges | Time steps | Sampling Rate | Window | Horizon |
|----------|------|-------|-------|------------|---------------|--------|---------|
| METR-LA | Directed | 207 | 1,515 | 34,272 | 5 minutes | 12 | 12 |
| PEMS-BAY | Directed | 325 | 2,369 | 52,128 | 5 minutes | 12 | 12 |
| EngRAD | Undirected | 487 | 2,297 | 26,304 | 1 hour | 24 | 6 |

**ROCKETMAN** This is the dataset used in the synthetic experiments in Sec. 5. We generate a graph with a given structure and $N = 16$ nodes. At each node, we generate a sequence of $T = 9$ time steps, again sampling values uniformly from the $[0, 1]$ interval. The task is to predict, for a target node $v$, the average value at time step $t - i$ of nodes located exactly $k$ hops away from $v$. That is, given a spatial distance $k \in [0, D]$, where $D$ is the graph's diameter, we define the source set $\mathcal{N}_k(v)$ as the nodes at shortest-path distance $k$ from $v$, with $\mathcal{N}_0(v) = \{v\}$. The label for node $v$ is then $\boldsymbol{y}_t^v = \sum_{u \in \mathcal{N}_k(v)} \frac{\boldsymbol{x}_{t-i}^u}{|\mathcal{N}_k(v)|}$, while predictions for all other nodes are masked out. We use the same train/validation/test split as in the other synthetic tasks. We use as graphs the RING and LOLLIPOP graphs illustrated in Fig. 6. We further show in Fig. 7 the performance of an MPTCN in this dataset with both TTS (7a) and T&S (7b) approaches.

We now present the real-world datasets used in the experiments of Tab. 1. We split datasets into windows of $T$ time steps, and train the models to predict the next $H$ observations. We closely follow the setup of previous works [40, 41]. In the following, we report detailed information for experiments on each dataset for completeness.

**METR-LA & PEMS-BAY** Both are widely popular benchmarks for graph-based spatiotemporal forecasting. PEMS-BAY contains 6 months of data from 325 traffic sensors in the San Francisco Bay Area, while METR-LA contains 4 months of analogous readings acquired from 207 detectors in the Los Angeles County Highway [19]. We use the same setup as previous works [41, 21, 19] for all the preprocessing steps. As such, we normalize the target variable to have zero mean and unit variance on the training set and obtain the adjacency matrix as a thresholded Gaussian kernel on the road distances [19]. We sequentially split the windows into $70\%/10\%/20\%$ partitions for training, validation, and testing, respectively. We use encodings of the time of day and day of the week as additional input to the model. For METR-LA, we impute the missing values with the last observed value and include a binary mask as an additional exogenous input. Window and horizon lengths are set as $T = 12$ and $H = 12$.

**EngRAD** The EngRAD dataset contains hourly measurements of 5 different weather variables collected at 487 grid points in England from 2018 to 2020. We use solar radiation as the target variable, while all other weather variables are used as additional inputs, along with encodings of the time of the day and the year. Window and horizon lengths are set as $T = 24$ and $H = 6$. We scale satellite radiation in the $[0, 1]$ range and normalize temperature values to have a zero mean and unit variance. We do not compute loss and metrics on time steps with zero radiance and follow the protocol of previous work [40] to obtain the graph and training/validation/testing folds.

**Training setting** We trained all models using the Adam [77] optimizer with an initial learning rate of $0.001$, scheduled by a cosine annealing strategy that decays the learning rate to $10^{-6}$ over the full training run. Gradients are clipped to a maximum norm of 5 to improve stability. For synthetic experiments, we trained for a maximum of 150 epochs with early stopping if the validation loss did not improve for 30 consecutive epochs, using mini-batches of size 32. To reduce computational time, we limit each epoch to the first 400 randomly sampled batches in the experiments for Fig. 4. For experiments on real-world datasets, we used the MAE as the loss function, trained for up to 200 epochs with a patience of 50 epochs, and in each epoch randomly sampled without replacement 300 mini-batches of size 64 from the training set.

**Baselines** In the following, we report the hyperparameters used in the experiment for the considered architectures. Whenever possible, we relied on code provided by the authors or available within open-source libraries to implement the baselines.

**MPTCN** We use $d = 64$ hidden units and the GELU activation function [78] throughout all layers. As MP, we use the Diffusion Convolution operator from [54]. For the real-world datasets, we compute messages with different weights from both incoming and outgoing neighbors up to 2 hops, as done by Li et al. [19]. As the ENCODER, we upscale the input features through a random (non-trained) semi-orthogonal $d_x \times d$ matrix, such that the norm of the input is preserved. As the READOUT, we use different linear projections for each time step in the forecasting horizon.

**GWNet** We used the same parameters reported in the original paper [21], except for those controlling the receptive field. Being GWNet a convolutional architecture, this was done to ensure that the receptive field covers the whole input sequence. In particular, we used 6 layers with

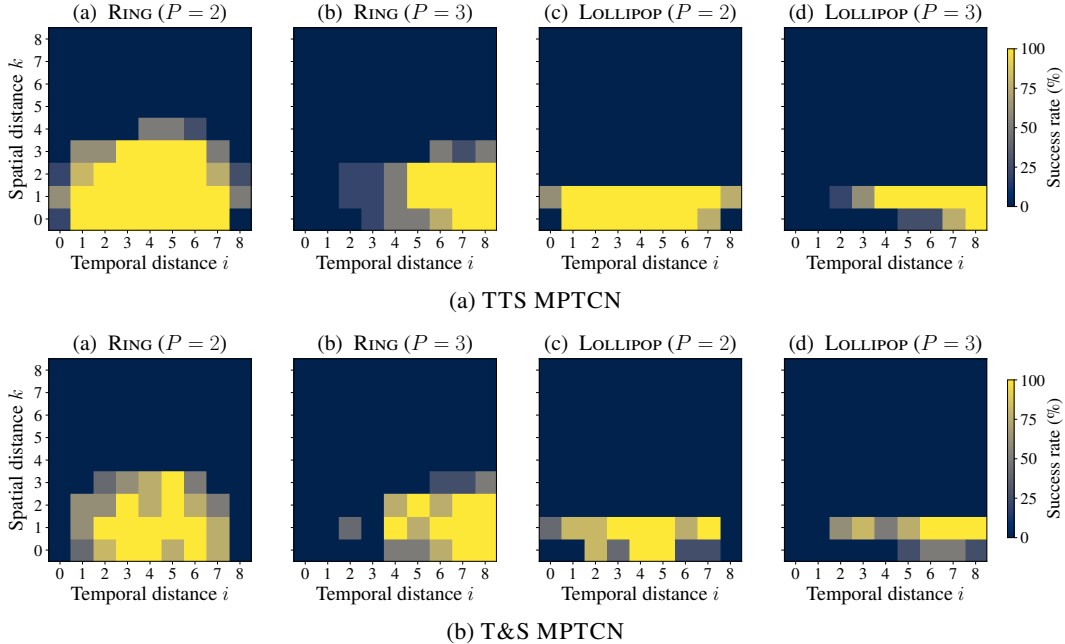

Figure 7: Success rate (%) of MPTCNs on the ROCKETMAN dataset. The tasks vary for the type of graph used (RING or LOLLIPOP) and size of $P$ (2 or 3). The plot axes show the source neighbors distance $k$ and the temporal distance $i$.

temporal kernel size and dilation of 3 for EngRAD since the input window has length 24. For the TTS implementation, we compute all message-passing operations stacked at the end of temporal processing, without interleaving any dropout or normalization (as done in the temporal part). The final representation is added through a non-trainable skip-connection to the output of temporal processing, to mimic the original T&S implementation. The two approaches share the same number of trainable parameters.

## C  Computational complexity

In this appendix, we analyze the computational complexity of MPTCNs with spatiotemporal message passing as defined in Eq. 8–10. A summary of computational complexities is provided in Tab. 4, where we omit constant factors and the dependency on feature dimension $d$ for clarity.

**Naive approach.**   We begin by analyzing the temporal component in Eq. 8, where the TMP operator is implemented as a stack of causal convolutional layers, each with kernel size $P$ as specified in Eq. 10. Each temporal layer performs $P$ matrix-vector multiplications and thus incurs a time complexity of $\mathcal{O}(Pd^2)$. With $L_\mathsf{T}$ such layers, the total cost of temporal processing becomes $\mathcal{O}(L_\mathsf{T}Pd^2)$ per node. Since this is applied independently to all $N$ nodes across $L$ outer layers, the cumulative temporal cost is $\mathcal{O}(LL_\mathsf{T}NPd^2)$. Next, we consider the spatial processing in Eq. 9, where the MP operator is composed of $L_\mathsf{S}$ message-passing layers, each defined as in Eq. 14. Assuming the message function $\phi^{(l)}$ involves a matrix multiplication, with complexity $\mathcal{O}(|\mathcal{E}|d^2)$, where $|\mathcal{E}|$ is the number of graph edges. Additionally, each layer applies two matrix-vector multiplications per node, one for $\Theta_\mathsf{U}^{(l)}$ and one for $\Theta_\mathsf{M}^{(l)}$. Hence, the cost of a single message-passing layer is $\mathcal{O}(|\mathcal{E}|d^2 + 2Nd^2)$. Repeating this over $L_\mathsf{S}$ spatial layers, $T$ time steps, and $L$ outer layers results in a total spatial cost of $\mathcal{O}(LL_\mathsf{S}T(|\mathcal{E}|d^2 + 2Nd^2))$. Combining both components, the overall computational complexity of an MPTCN with number of layers $L$, $L_\mathsf{T}$, and $L_\mathsf{S}$, kernel size $P$, and hidden dimension $d$ is:

$$\mathcal{O}\left(L\left(L_\mathsf{T}NPd^2 + L_\mathsf{S}T(|\mathcal{E}|d^2 + 2Nd^2)\right)\right).$$

**Optimized approach.**   Encoder-decoder architectures, as the STGNNs under study, typically require only the final representation $\boldsymbol{h}_t^{v(L)}$ to produce the output $\hat{\boldsymbol{y}}_t^v$. As a result, the last STMP layer only

Table 4: Comparison of computational complexity between TTS and T&S under fixed spatial and temporal budgets ($B_S = LL_{\mathsf{S}}$, $B_T = LL_{\mathsf{T}}$), and fixed kernel size $P$, assuming that each layer's receptive field satisfies $PL_{\mathsf{T}} \geq T$. The TTS approach achieves a $T$-fold reduction in computation.

| | TTS (L=1) | T&S (L>1) |
|---|---|---|
| **Naive** | $\mathcal{O}\left(B_T NP + B_S T|\mathcal{E}|\right)$ | $\mathcal{O}\left(B_T NP + B_S T|\mathcal{E}|\right)$ |
| **Optimized** | $\mathcal{O}\left(B_T NP + B_S|\mathcal{E}|\right)$ | $\mathcal{O}\left(B_T NP + L_{\mathsf{S}}|\mathcal{E}| + (B_S - L_{\mathsf{S}})T|\mathcal{E}|\right)$ |

requires performing message passing exclusively at the last time step $t$, using the embeddings $\boldsymbol{H}_t^{(L)}$. This reduces the spatial computation at the final layer to $\mathcal{O}\left(L_{\mathsf{S}}(|\mathcal{E}|d^2 + 2Nd^2)\right)$, which is a factor of $T$ more efficient than in the naive approach. Assuming that $PL_{\mathsf{T}} > T$, i.e., each STMP layer has a temporal receptive field that spans the entire sequence, all preceding STMP layers still require access to all $T$ time steps and therefore cannot be similarly optimized. Under this assumption, the total complexity becomes:

$$\mathcal{O}\left(LL_{\mathsf{T}}NPd^2 + ((L-1)L_{\mathsf{S}}T + L_{\mathsf{S}})\left(|\mathcal{E}|d^2 + 2Nd^2\right)\right),$$

and simplifies in the case of a TTS architecture with $L = 1$ to $\mathcal{O}\left(L_{\mathsf{T}}NPd^2 + L_{\mathsf{S}}(|\mathcal{E}|d^2 + 2Nd^2)\right)$. If we relax the assumption $PL_{\mathsf{T}} > T$, the T&S method still yields a computational gain over the naive implementation for the layers required to cover the full sequence length $T$. The remaining layers, however, incur the same cost as in the naive case. This results in a moderate speedup, albeit still significantly less efficient than the TTS approach.

# D    TEMPORALNEIGHBOURSMATCH

We propose TEMPORALNEIGHBOURSMATCH, an adaptation of NEIGHBOURSMATCH [30] to the spatiotemporal setting. In NEIGHBOURSMATCH, information is propagated from sender nodes to a root node, and the goal is to classify the root node with the label of the sender node with matching features. We extend this to a spatiotemporal setting with fixed graph topology in time and a single active time step, where only sender nodes receive non-zero features. This time step is an additional hyperparameter, akin to depth in TREENEIGHBOURSMATCH, where the graph topology is a tree. The goal in the TEMPORALNEIGHBOURSMATCH problem remains to route the correct sender's label – identified by matching features – to the root node at a later step. This task complements our existing synthetic benchmarks, as it emphasizes information compression, i.e., the need to retain and route specific input signals through bottlenecks.

For our experiment, we use a TEMPORALTREENEIGHBOURSMATCH controlled environment, in which the topology is a tree and is fixed, while the nonzero leaf features are placed either at the initial or the final time step. In Tab. 5, we report test accuracy in the range $[0, 1]$ for varying tree depths and a fixed number of time steps $T = 4$. Regarding spatial over-squashing, we observe the same pattern as in the original experiments of Alon and Yahav [30]. Indeed, accuracy begins to drop at depth 4 and shows substantial degradation at depth 5. Notably, performance is higher at depth 4 when the relevant information appears at the start of the sequence rather than at the end, consistent with our theoretical and empirical findings. However, the large drop in accuracy at depth 5 in both scenarios suggests that the main bottleneck is spatial rather than temporal. This indicates that the task, when combined with a tree-like topology, is limited in its ability to capture the nuances of the spatiotemporal bottleneck problem.

Table 5: Test accuracy on TEMPORALTREENEIGHBOURSMATCH with varying tree depth and feature position.

| | TREE DEPTH | | |
|---|---|---|---|
| **FEATURES POSITION** | 3 | 4 | 5 |
| **First step** | 1.00 ±0.00 | 1.00 ±0.00 | 0.09 ±0.01 |
| **Last step** | 1.00 ±0.01 | 0.73 ±0.46 | 0.11 ±0.08 |

