# OpenReview forum: "Over-squashing in Spatiotemporal Graph Neural Networks"
_NeurIPS.cc/2025/Conference — NeurIPS 2025 poster_

### Official Review · Reviewer_sWQM · 2025-06-04

**Clarity:** 3
**Significance:** 2
**Originality:** 2
**Rating:** 4
**Confidence:** 4

**Summary:**

This paper investigates the over-squashing issue in Spatiotemporal Graph Neural Networks (STGNNs), where information from distant nodes and time steps fails to propagate effectively due to bottlenecks.

The authors formalize this spatiotemporal over-squashing problem, demonstrating its distinct characteristics from the static case, including a counterintuitive finding that convolutional STGNNs can favor temporally distant information.

Their theoretical and empirical results show that different architectural paradigms (time-and-space vs. time-then-space) are equally affected, providing justification for computationally efficient designs and offering principled guidance for more effective STGNN architectures.

**Questions:**

1. The synthetic tasks (COPYFIRST/LAST, k-hop average) are illustrative but do not fully capture the essence of spatiotemporal bottleneck challenges in the same fundamental way as NeighboursMatch in prior work [3]. Could the authors discuss the challenges in designing an analogous 'Spatiotemporal NeighboursMatch' or a similar fundamental task? What would be the key characteristics of such a task (e.g., requiring precise information from specific spatiotemporal locations across bottlenecks)? Even if not fully implemented, a conceptual discussion could be valuable.
2. This work primarily focuses on temporal rewiring (RD, RN) to address temporal over-squashing. Theorem 5.1 suggests that spatiotemporal over-squashing arises from the combined effects. What would the interplay be between the proposed temporal rewiring strategies and existing spatial graph rewiring techniques? Would applying both be additive, or are there more complex interactions?
3. Theorem 5.1 provides a compelling theoretical argument for the similar impact of over-squashing on TTS and T&S architectures under the lens of the derived Jacobian bound. However, practical performance can also be influenced by optimization landscapes, expressivity differences not captured by this specific bound, or sensitivity to hyperparameter choices. Could the authors elaborate on the extent to which the empirical results (Table 1, and perhaps unreported ablations) support this equivalence more broadly?

**Ethical Concerns:**

["NO or VERY MINOR ethics concerns only"]

**Final Justification:**

After considering the paper, the rebuttal, and discussions, I maintain a borderline recommendation, leaning slightly positive due to the paper's clear theoretical framing and timely focus.

Suggestion: Please include a strong empirical benchmark similar to NeighboursMatch in the final version to better validate the theoretical claims. A preliminary evaluation of recurrent architectures on this task, along with a more in-depth analysis of convolutional models, would significantly enhance the practical impact of the work.

**Limitations:**

Adequately addressed.

**Paper Formatting Concerns:**

No major issues

**Quality:**

2

**Strengths And Weaknesses:**

## Strengths

1. The paper provides a theoretical foundation for its claims. It formalizes spatiotemporal over-squashing using Jacobian-based sensitivity analysis (Eq. 11, 12) and derives explicit bounds (Theorem 4.1, Theorem 5.1) that clearly separate model parameter contributions from topological contributions (both spatial S and temporal R).
2. The paper is overall well written, with a clear structure that outlines the problem, contributions, and methodology in a logical flow. It uses consistent notation, helpful visualisations, and intuitive explanations to convey over-squashing, making the technical content accessible.

$~$

## Weaknesses

1. The paper primarily focuses on STGNNs where the temporal component is a Temporal Convolutional Network (TCN). While TCNs are a valid and increasingly popular choice, many STGNNs also employ Recurrent Neural Networks (RNNs) for temporal modeling [1, 2]. The analysis might not directly translate to RNN-based STGNNs, a point acknowledged in the Limitations and future work section but important for contextualizing the paper's scope.
2.  The issue of over-squashing was clearly demonstrated in static GNNs through the fundamental NeighboursMatch task introduced in prior work [3]. Multiple GNNs were shown to suffer from the issue. However, this paper does not provide any compelling fundamental benchmarks to assess over-squashing in STGNNs.
3. The core ideas of over-squashing, sensitivity analysis via Jacobians, and graph rewiring are established concepts in the GNN literature, restricting the originality of the work.

$~$

References:
1. Spatio-Temporal Graph Neural Networks: A Survey, In arXiv 2301.10569.
2. Graph Neural Networks for Temporal Graphs: State of the Art, Open Challenges, and Opportunities, In TMLR 2024.
3. On the Bottleneck of Graph Neural Networks and its Practical Implications, In ICLR 2021.

---

> ### Author Rebuttal · Authors · 2025-07-31
>
> We thank the reviewer for the constructive feedback and for recognizing the clarity of our theoretical framework and the quality of the visualizations. Below, we address the concerns raised regarding scope, evaluation methodology, and originality.
>
> 1. **W1. Extension to RNN-based STGNNs**
>    We agree that extending the analysis to RNN-based STGNNs is an important and valuable direction. In this work, we focus on convolutional STGNNs (specifically MPTCNs) due to their widespread use and because their architectural structure lends itself to tractable theoretical analysis. In particular, the analogy between temporal and spatial convolutions enables a unified sensitivity framework. RNNs, by contrast, involve sequential dynamics with nonlinear hidden-state recursion, which introduce fundamentally different analytical challenges. A unified treatment would require distinct tools and significantly broaden the scope beyond what is feasible in a single paper. We view our work as a foundation for such future extensions and have clarified this in the conclusion.
> 2. **W2 & Q1. Designing a Spatiotemporal NeighboursMatch to Complement Propagation Benchmarks**
>    We thank the reviewer for this excellent suggestion. We agree that a spatiotemporal NeighboursMatch would be a valuable addition to our evaluation suite, as it would evaluate complementary aspects to over-squashing.
>
>    We propose **TemporalNeighboursMatch**, an adaptation of NeighboursMatch \[1\] to the spatiotemporal setting. In NeighboursMatch, information is propagated from *sender* nodes to a root node, with the correct sender identified by matching node degrees to the root node. We extend this to a temporal setting with fixed graph topology in time and a single **active time step**, where only sender nodes receive non-zero features. This time step is a hyperparameter, akin to depth in TreeNeighboursMatch. The goal remains to route the correct sender’s feature—identified by matching degree—to the root at a later step.
>
>    We are currently implementing this synthetic benchmark and plan to run experiments during the rebuttal period. While results are not yet available, we believe it will meaningfully strengthen our evaluation, and we will report updates as they become available.
>
>
>
>    This proposed task complements our existing benchmarks. COPYFIRST and COPYLAST test **temporal memory and long-range propagation**, while RocketMan (k-hop average) targets **spatiotemporal information propagation bottlenecks**, extending prior synthetic tasks such as **GraphTransfer \[2\]** and **RingTransfer \[3\]**. In particular, it requires retrieving information from a specific spatiotemporal location, with temporal and spatial distances, as well as graph topology, jointly configurable. In contrast, NeighboursMatch-style tasks emphasize **information compression**—the need to retain and route specific input signals through bottlenecks. Together, these benchmarks help capture multiple aspects of over-squashing in spatiotemporal models.
>
>
>
> 3. **W3. Originality of the work compared to GNN literature**
>    Our main contribution is to extend over-squashing analysis, traditionally studied in static GNNs, to the **spatiotemporal setting**, where temporal structure interacts with graph topology in nontrivial ways. Prior work has not addressed how over-squashing manifests when information must propagate both across nodes and over time. While tools like Jacobian sensitivity and rewiring are established in the static case, applying them to STGNNs introduces new challenges. For example, temporal message passing reshapes receptive fields, and static formulations do not capture how temporal and spatial dependencies interact. Our theoretical results (Theorems 4.1 and 5.1) explicitly disentangle these effects, and our empirical findings uncover behaviors unique to the temporal dimension, such as non-monotonic sensitivity with respect to time. Although STGNNs build on GNN components, their information flow is structurally different. We believe that adapting core techniques to this setting, and revealing new failure modes, adds both novelty and value.
> 4. **Q2. Interplay of temporal and spatial rewiring**
>    This is an insightful question. As Theorem 5.1 shows, temporal and spatial topologies jointly influence the sensitivity through a multiplicative relationship. Improving only one component (e.g., temporal connectivity) may not mitigate over-squashing if the other remains bottlenecked. We conducted additional experiments combining temporal and spatial rewiring, through row-normalized convolutions and FoSR, to assess their combined effect. Please see **answer n.4** in the rebuttal for reviewer **QFNp**.
> 5. **Q3. Empirical support for the equivalence of TTS and T\&S models**
>    We agree with the reviewer that, beyond Jacobian-based sensitivity bounds, forecasting accuracy can also be influenced by optimization dynamics, model expressivity, and hyperparameter sensitivity. While Theorem 5.1 shows that TTS and T\&S architectures face similar over-squashing risks in theory, this does not imply identical empirical performance. In our experimental setup, TTS and T\&S models are deliberately designed to be **highly comparable**, differing only in the order in which spatial and temporal processing is applied while keeping the same temporal and spatial budgets. When trained under identical hyperparameter settings, we observe that TTS models tend to perform slightly better across both synthetic and real-world datasets. However, we acknowledge that T\&S architectures may benefit from different optimization choices (e.g., learning rate, initialization, or depth), which we have not extensively tuned. The observed performance gap is modest, but we agree that a more thorough empirical analysis—particularly examining optimization dynamics—would further complement the theoretical findings. We consider this a valuable direction for future work.
>
> We appreciate the reviewer’s thoughtful feedback and hope our clarifications help contextualize the scope and novelty of our contributions. We would be grateful if these responses support a more favorable evaluation.
>
> ---
>
> [1] Alon, U., & Yahav, E. “On the bottleneck of graph neural networks and its practical implications.” International conference on Learning Representations, 2021
> [2] Di Giovanni et al. "On over-squashing in message passing neural networks: The impact of width, depth, and topology." International conference on machine learning. PMLR, 2023\.
> [3] Gravina et al. "On oversquashing in graph neural networks through the lens of dynamical systems." Proceedings of the AAAI Conference on Artificial Intelligence. Vol. 39\. No. 16\. 2025\.

---

> > ### Author Response · Authors · 2025-08-08
> >
> > As a follow-up to our definition of the **TemporalNeighboursMatch** problem in the previous rebuttal, we present preliminary experimental results. In the following table, we report test accuracy (in the range \[0,1\]) for varying tree depths under two conditions: nonzero leaf features placed either at the initial or the final time step.
> >
> > | TREE DEPTH | Features in LAST time step | Features in FIRST time step |
> > | :---: | :---: | :---: |
> > | 3 | 1.00±0.01 | 1.00±0.00 |
> > | 4 | 0.73±0.46 | 1.00±0.00 |
> > | 5 | 0.11±0.08 | 0.09±0.01 |
> >
> > Regarding **spatial over-squashing**, we observe the same pattern as in the original experiments of \[1\]: accuracy begins to drop at depth 4 and shows substantial degradation at depth 5\. Notably, at depth 4, performance is higher when the relevant information appears at the **start** of the sequence rather than at the **end**, consistent with our theoretical and empirical findings.
> >
> > We agree this task is of significant interest and intend to explore it in greater depth, providing an expanded set of results in the revised manuscript. We hope these additional experiments help clarify our position and are happy to elaborate further should any questions remain.
> >
> > ---
> > [1] Alon, U., & Yahav, E. “On the bottleneck of graph neural networks and its practical implications.” International conference on Learning Representations, 2021

---

### Official Review · Reviewer_QFNp · 2025-06-25

**Clarity:** 4
**Significance:** 2
**Originality:** 3
**Rating:** 4
**Confidence:** 3

**Summary:**

This paper investigates the phenomenon of over-squashing in Spatiotemporal Graph Neural Networks (ST-GNNs), where long-range spatial and temporal dependencies are inadequately propagated due to the bottlenecking of information in message passing. The authors theoretically characterize how spatiotemporal over-squashing arises from the interaction between graph topology and temporal resolution, extending prior over-squashing definitions from static GNNs.  Experiments on tailored synthetic memory tasks (Fig. 2–4) and on three traffic-style forecasting benchmarks (METR-LA, PEMS-BAY, EngRAD) confirm the theory and show that row-normalised filters and TTS variants either match or slightly outperform their vanilla or T&S counterparts while being more scalable

**Questions:**

1. It is clear that the sensitivity is presented by upper-bound, this upper-bound does not offer much in understanding over-squashing issues.  Ideally we shall have a lower-bound.    I understand that this result from [42].  My question here is how this upper-bound can benefit analysis.

2. Author(s) may consider comparing row-normalised MPTCN (TTS) with at least one transformer-style STGNN (e.g., StemGNN or CrossFormer) and with a graph-rewiring baseline such as FoSR or Locality-Aware Rewiring on METR-LA. Report MAE and inference latency.

3. Can the Jacobian factorisation be extended to a joint 2-D convolution (shared kernels over space & time)? A short appendix note or experiment would strengthen the theoretical scope.

**Ethical Concerns:**

["NO or VERY MINOR ethics concerns only"]

**Final Justification:**

Thank you the author(s) for your clarification on my previous misunderstanding, taking my suggestion into account and completing several extra experiments which can be revised into the new version of the paper. I have no more questions, and at this stage I support the acceptance of the paper.

**Limitations:**

The paper lacks a direct discussion of its limitations.

**Paper Formatting Concerns:**

None noted. The paper follows NeurIPS 2025 formatting guidelines.

**Quality:**

3

**Strengths And Weaknesses:**

Strengths:

1.	First systematic theory of over-squashing in spatiotemporal GNNs; the factorised Jacobian bound (Theorem 5.1) extends static-graph analyses to a two-axis setting and immediately explains several empirical quirks (sink bias, depth brittleness).

2.	The paper extends the definition of over-squashing by formulating a “temporal effective resistance” metric that captures how temporal bottlenecks compound spatial ones.

3.  The paper is clearly written.   The paper is modular (Sec. 4  temporal, Sec. 5 full STGNN), proofs are deferred yet sketched, and a separate limitations paragraph is included.

Weaknesses:

1.	While the definition of temporal effective resistance is interesting, the results can be implied from the existing given by Di Giovanni et al. [42], in fact MPTCN is a combination of two GNN operations.

2. Real-data experiments compare only within the MPTCN family (vanilla vs RN vs Dilation) and Graph WaveNet; no transformer-based STGNNs, diffusion-imputation models or recent graph-rewiring baselines are included, limiting claims of generality.

---

> ### Author Rebuttal · Authors · 2025-07-31
>
> We thank the reviewer for their thoughtful feedback and for highlighting the strengths of our work, particularly the theoretical characterization of spatiotemporal over-squashing and the clarity of our exposition. Below, we respond to the main concerns and questions raised.
>
> 1. **W1. Relation to results on GNNs**
>    We agree that temporal and spatial convolutions share structural similarities, and this motivated our decision to focus on convolutional STGNNs. However, as we highlight in Theorem 4.1, the temporal convolutional setting exhibits specific behaviors not directly implied by the more general spatial (GNN) case. In particular, the causal and sequential structure of temporal topology introduces new forms of bottlenecking not directly applicable to arbitrary graphs. This distinction allows us to derive sharper theoretical insights that are specific to the spatiotemporal setting and not straightforward extensions of existing GNN results. We believe this makes the formalization meaningful, novel, and distinct.
> 2. **W2. Comparison with non-convolutional models**
>    Our experimental objective was not to comprehensively benchmark all STGNN architectures, but to validate our theoretical results for MPTCNs on not only synthetic tasks but also real-world settings, an aspect often overlooked in theoretical studies. We included Graph WaveNet, a still-competitive and widely used STGNN, to demonstrate that the observed similarities between TTS and T\&S architectures concerning over-squashing in factorized convolutional STGNNs also hold in more complex settings that incorporate additional components such as dropout, batch normalization, and residual connections. **We explicitly avoid making claims of generality** beyond this class of models. Our assumptions and scope are stated clearly throughout the paper. Nonetheless, we appreciate the suggestion and plan to extend our comparisons in future work to include other common STGNN designs.
> 3. **Q1. Significance of upper-bound analysis**
>    We appreciate this important question. While upper bounds may not fully capture the tightest sensitivity behavior, they serve a critical role in identifying potential bottlenecks. Specifically, the upper bound in Theorem 4.1 reveals how variations in the temporal topology matrix $R$ can limit the propagation of information between time steps. For instance, a small $(R)\_{ij}$ results in reduced sensitivity from $t-i$ to $t-j$. We corroborate this behaviour empirically in our synthetic experiments, demonstrating that the bound, though conservative, provides practical guidance in identifying over-squashing. We fully agree that developing a lower bound would yield a valuable yet complementary perspective; however, it remains an open problem, and we regard this as a promising direction for future work. We will add a note on the limitation of the upper bound in the limitations section.
> 4. **Q2. Comparison with graph-rewiring baselines**
>    While the primary goal of our experimental section was to validate our theoretical results within the MPTCN family, we recognize the value of broader comparisons, particularly incorporating graph rewiring components alongside our proposed temporal mitigations. As suggested by the reviewer, we adopted FoSR \[1\] as the graph rewiring method and evaluated its performance with and without row-normalized convolutions on the EngRAD dataset. We selected EngRAD due to its symmetric topology, which makes rewiring more meaningful compared to traffic forecasting tasks, where spatial structure is rigidly defined by the underlying road network. Besides our original MPTCN implementation relying on Diffusion Convolution (DCNN) as message-passing operation, we further consider RGCN \[2\], to weight differently the contribution of rewired edges. We report results in terms of MAE in the TTS setting below:
>
>    | MP operation   | $R$            | $R_N$          |
>    |----------------|----------------|----------------|
>    | DCNN           | 43.50 ± 0.08   | 40.30 ± 0.16   |
>    | RGCN           | 43.78 ± 0.29   | 41.10 ± 0.11   |
>    | Original       | 44.47 ± 0.42   | 40.38 ± 0.08   |
>
>    These results indicate that combining both spatial and temporal rewiring yields the best average performance when using DCNN. In particular, each technique individually improves performance, with temporal rewiring contributing the largest marginal gain. This is consistent with our theoretical analysis, reinforcing that temporal bottlenecks are a significant limiting factor in STGNNs.
>    We also measured inference latency in terms of batches processed per second at test time. We found that spatial rewiring introduces a substantial overhead (\~70% throughput compared to models without rewiring), whereas row normalization has only a minor effect (\~92%). These findings will be included in the revised version of the paper to address this concern more thoroughly.
> 5. **Q3. Extension to joint space-time convolution**
>    Thank you for this insightful suggestion. Our current results are derived under a factorized architecture, which constrains the Jacobian structure in a way that allows clean theoretical bounds and comparable experimental frameworks, applicable to both TTS and T\&S variants. Joint space-time convolution (e.g., 2D kernels over space-time) represents a broader class, which enables interactions among different nodes and time steps spanning both dimensions simultaneously. In this case, the factorization is not possible unless we assume these cross-dimensional interactions to be blocked, resorting to our factorized case. Generalizing the sensitivity analysis to arbitrary joint convolutions is non-trivial due to the coupling of spatial and temporal dependencies, but we agree that this is a promising direction. We have added a note on this in the discussion section and plan to explore this extension in future work.
> 6. **Discussion of limitations**
>    > "The paper lacks a direct discussion of its limitations."
>
>     We do include a limitations paragraph in Section 7, where we state that our theoretical framework applies primarily to convolutional architectures, and does not cover recurrent or general joint spatiotemporal models. We will revise this section to more explicitly highlight the limits of our analysis and assumptions to ensure clarity.
>
> We appreciate the reviewer’s constructive feedback and hope that our responses and additions help clarify the paper’s scope and contributions. We would be grateful if these points might support a more favorable reassessment.
>
> ---
> [1] Karhadkar et al. "FoSR: First-order spectral rewiring for addressing oversquashing in GNNs." arXiv preprint arXiv:2210.11790 (2022).
> [2] Schlichtkrull et al. "Modeling relational data with graph convolutional networks." European semantic web conference. Cham: Springer International Publishing, 2018.

---

> > ### Comment · Reviewer_QFNp · 2025-07-31
> > **Thanks**
> >
> > Thank you the author(s) for your clarification, taking my suggestion into account and completing several extra experiments which can be revised into the new version of the paper. I have no more questions, and at this stage I support the acceptance of the paper.

---

> > > ### Author Response · Authors · 2025-08-07
> > >
> > > We are pleased to have addressed the reviewer’s concerns and appreciate their positive reassessment. The new experiments will be incorporated into the revised manuscript. We are grateful for the reviewer’s time, suggestions, and valuable feedback provided throughout the review process.

---

### Official Review · Reviewer_c2D6 · 2025-06-30

**Clarity:** 4
**Significance:** 4
**Originality:** 3
**Rating:** 5
**Confidence:** 4

**Summary:**

This paper introduces a formal characterization of over-squashing in Spatiotemporal Graph Neural Networks (STGNNs). It highlights how the temporal dimension amplifies information compression, distinguishing it from static graph over-squashing. The authors demonstrate that convolutional STGNNs counterintuitively favor information propagation from temporally distant points. They prove that both time-and-space (T&S) and time-then-space (TTS) processing paradigms are equally affected by this phenomenon, providing theoretical justification for computationally efficient TTS implementations. The paper validates these findings on synthetic and real-world datasets, offering insights for more effective STGNN designs.

**Questions:**

1.  Can you further explain why your causal convolution setup reduces sensitivity to recent information? This seems contrary to typical causal convolution properties. How might this impact tasks highly dependent on very recent history?

2.  For dilated convolutions, how can we best choose parameters like $P$ and $M$ to avoid reintroducing over-squashing? Can you elaborate on the specific limitations of row-normalization for STGNN tasks beyond forecasting?

3.  Your claims on TTS vs. T&S equivalence apply to MPTCNs. Do you expect this equivalence to hold for recurrent STGNNs or models with joint space-time filters? I am interested if you you think these results will be generalizable across architectures as I dont seem to find any part of the theorems which are architecture specific

I understand this is mainly a theoretical paper but I felt the experimental section of the paper to be insufficient to empirically prove the claims the authors made in the theoretical sections.

I would be very interested if you could conduct a synthetic sensitivity analysis for the parameters of the proposed mitigation strategies. For dilated convolutions ($R_D$), systematically vary $P$  and $M$ to quantify how they influence the reintroduction of over-squashing patterns (as seen in Figure 2 (b.2) ). For row-normalization ($R_N$), explore its effects on tasks where non-final time steps are crucial, to quantify its specific limitations beyond forecasting.

I think this will add a lot of value to the paper

**Ethical Concerns:**

["NO or VERY MINOR ethics concerns only"]

**Final Justification:**

I will maintain my positive score for this paper. The authors provided a thorough and thoughtful rebuttal that successfully addressed my concerns. One key thing is that have committed to making key changes in the final version, including clearer explanations, improved visualizations, and a deeper discussion of the nuances and limitations of their mitigation strategies.

The paper's core contributions—a novel theoretical framework for over-squashing in STGNNs and the practical insights derived from it—remain strong and compelling.

**Limitations:**

yes

**Quality:**

3

**Strengths And Weaknesses:**

### Strengths

1. The paper introduces the concept of over-squashing specifically for Spatiotemporal Graph Neural Networks (STGNNs), a previously unexplored area despite its importance in static GNNs. This is a critical gap addressed. The formal characterization of spatiotemporal over-squashing is a novel and significant theoretical contribution
2. The theoretical framework is robust. It formally characterizes spatiotemporal over-squashing and provides clear bounds on information propagation for TCNs and MPTCNs. The paper effectively disentangles the spatial and temporal components of over-squashing, which is crucial for understanding STGNNs. The finding that Time-then-Space (TTS) and Time-and-Space (T&S) paradigms are equally affected by over-squashing, despite TTS models offering computational benefits, is a key theoretical insight for practical design.
3. The theoretical claims are strongly backed by comprehensive empirical evaluations on both synthetic and real-world datasets (METR-LA, PEMS-BAY, EngRAD). The synthetic memory tasks are well-designed to isolate and demonstrate spatiotemporal bottlenecks. The consistent performance of TTS and T&S models in experiments reinforces the theoretical arguments about information bottlenecks.
4. The paper's counterintuitive discovery that convolutional STGNNs favor information propagation from temporally distant points is a significant finding. This "temporal over-squashing," likened to the attention sink effect in Transformers, offers crucial insights into TCN behavior.

### Weaknesses

 1. The primary focus is on "factorized convolutional approaches" (MPTCNs). While relevant, the analysis explicitly excludes recurrent architectures and models with joint space-time filters. This limits the generalizability of the theoretical equivalence between TTS and T&S to the broader STGNN landscape.

2. The statement that causal convolutions "progressively diminish sensitivity to recent information" and "undermine the locality bias"  needs more detailed explanation. This appears counterintuitive to the common understanding of causal convolutions' role in preserving temporal locality. A deeper dive into the specific mechanism leading to this "inverted pattern" within their formulation would enhance clarity.

3. While dilated convolutions and row-normalization are proposed, the discussion on their limitations could be expanded. For instance, the reintroduction of over-squashing with dilated convolution resets (Fig. 2 b.2) warrants a more thorough analysis of optimal parameter selection to mitigate this issue. Similarly, the benefits of row-normalization primarily for forecasting tasks suggest potential limitations for other STGNN applications that are not fully explored.

 Minor Visual Clarity in Figure 2: While the text clearly describes the behavior of $R_N$, the visual representation in Figure 2 (c.1 & c.2) for row-normalized convolutions could be slightly clearer in depicting the convergence to a uniform distribution on the first column, as stated in the text and Proposition A.2.

---

> ### Author Rebuttal · Authors · 2025-07-31
>
> We sincerely thank the reviewer for the thoughtful and encouraging feedback. We are glad that the reviewer recognizes the novelty, significance, and robustness of our theoretical framework and its practical relevance. Below, we address each of the raised points.
>
> 1. **W1 & Q3. Extension to recurrent architectures and models with joint space-time filters**
>    We fully agree that extending the analysis to recurrent architectures or models with joint space-time filters is a valuable direction for future work. In this paper, we focused on delivering a rigorous and targeted analysis of spatiotemporal over-squashing within a widely-used class of factorized convolutional STGNNs.
>    Recurrent STGNNs following a T\&S paradigm exhibit fundamentally different information propagation dynamics. In these models, the spatial receptive field typically expands with the sequence length, differently from the TTS counterpart, where the entire temporal sequence is encoded into a latent state later used for message passing. A thorough theoretical analysis of recurrent STGNNs would require substantially broader analysis, which we view as out of scope for a conference-length submission.
>    Conversely, joint space-time filters are more closely aligned with our setting. The factorized convolutional models we study can be viewed as a special case of joint space-time architectures with a constrained parameterization. Therefore, our theoretical results apply directly to a subset of joint models where cross-dimensional edges are not considered. However, generalizing our sensitivity bounds to arbitrary joint space-time filters – which by design follow a T\&S scheme only – would require analyzing a larger architectural space. We consider this a promising future direction and have added a note on this in the discussion.
> 2. **W2 & Q1. Reduced sensitivity towards earlier time steps**
>    We agree that the finding appears counterintuitive at first glance. Our analysis shows that this effect emerges at depth, rather than in shallow causal convolutional networks. Specifically, Proposition 4.2 shows that as more convolutional layers are stacked, the influence of temporally recent inputs diminishes relative to more distant ones. This behavior stems from the structure of causal convolutions, which incrementally incorporate more information into a fixed-length context vector. When viewed through the matrix-multiplication perspective (i.e., using the Toeplitz matrix representation), it becomes evident that causal convolutions propagate information along powers of a directed path graph. Over multiple layers, earlier time steps accumulate influence through an increasing number of propagation paths, while more recent inputs have fewer paths for propagating their initial information. Importantly, because causal convolutions are forward-only, each time step can preserve its information in the associated context vector through self-loops only, with a major impact on the last time step in the sequence. When a task requires preserving local or recent information, this primacy bias can negatively impact performance even in trained neural networks, as demonstrated in the CopyLast experiment (Fig. 3). We have expanded the relevant discussion to clarify this mechanism and its practical implications for tasks that rely heavily on recent inputs.
> 3. **W3 & Q2. Limitations of mitigation strategies**
>    Thank you for highlighting the limitations of the proposed mitigation techniques.
>     - **Dilated convolutions ($R\_D$):** The parameters $P$ (dilation base) and $m$ (reset modulo) are primarily chosen to control the temporal receptive field at each layer. To avoid the sink effect (seen in Fig. 2(b.2)), we require $m \> L$, i.e., no resets within the model depth. However, when the dilation rate grows beyond the window size at layer $l \> \\log\_P T$, the convolution degenerates into a fully-connected layer shared across time. In these cases, resets become necessary for temporal propagation, but reintroduce the risk of over-squashing. Smaller $P$ allows more flexibility in setting $m$ before hitting this tradeoff.
>
>     - **Row-normalization ($R\_N$):** While effective in retaining local information for the last time step – useful for encoder-decoder architectures – the behavior for earlier steps is still similar to the vanilla model. As shown in Fig. 2 (c.2), however, paths toward the final time step dominate compared to any other paths in the temporal graph. Thus, for tasks requiring readout at intermediate time steps (e.g., imputation), the benefits of $R\_N$ may be limited. We have added a clarifying note in the discussion of mitigation strategies.
>
> We appreciate the suggestion to conduct a systematic sensitivity analysis of the mitigation strategies. While we agree that this could yield interesting insights, we believe it would fall outside the intended scope and focus of this work. As the reviewer kindly acknowledged, our paper is primarily theoretical, and our experiments were designed to validate core theoretical claims rather than to fully optimize or benchmark each mitigation strategy. We have added a brief mention of this limitation and encourage future work to build upon these mitigation techniques with a more exhaustive empirical lens.

---

> > ### Comment · Reviewer_c2D6 · 2025-08-05
> >
> > Thank you for your detailed rebuttal. I appreciate the clarifications and believe you’ve addressed most of my concerns. A couple of final minor requests:
> >
> > 1. Please add a brief illustrative example or visualization (e.g., sensitivity curves for a 3-layer vs. 6-layer TCN) to make the counterintuitive effect more tangible.
> >
> > 2. Explicitly annotate the heatmap axes and include a color-scale legend highlighting key sensitivity thresholds (e.g., “high,” “medium,” “low”).
> >
> > With those additions, I’m satisfied that all my points are addressed.

---

> > > ### Author Response · Authors · 2025-08-07
> > >
> > > We are glad the clarifications helped address the reviewer’s concerns. We will incorporate the two suggestions (sensitivity plots and heatmaps annotations) in the revised manuscript, as we agree that they would improve clarity and accessibility. We thank the reviewer for engaging in the discussion period and for the constructive feedback.

---

### Official Review · Reviewer_7S5V · 2025-07-02

**Clarity:** 3
**Significance:** 3
**Originality:** 3
**Rating:** 5
**Confidence:** 3

**Summary:**

This paper investigates over-squashing in Spatiotemporal Graph Neural Networks (STGNNs), where information from distant nodes is overly compressed. It formalizes the issue in the spatiotemporal setting and shows that causal convolutions can unintentionally increase sensitivity to distant inputs. Both time-and-space (T&S) and time-then-space (TTS) architectures are equally affected, indicating TTS’s efficiency doesn’t worsen bottlenecks. Results on synthetic and real-world tasks offer insights for alleviating information loss in STGNNs.

**Questions:**

See Weakness part.

**Ethical Concerns:**

["NO or VERY MINOR ethics concerns only"]

**Final Justification:**

I maintain my positive score.

**Limitations:**

See Weakness part.

**Quality:**

3

**Strengths And Weaknesses:**

### Strengths
1. The paper tackles an underexplored yet important problem in STGNNs: over-squashing in joint spatiotemporal networks, which has received little rigorous analysis in prior work.
2. The mathematical formalization of spatiotemporal over-squashing is well-structured, offering a strong theoretical foundation (Sections 4 and 5).
3. The synthetic experiments effectively validate the theory and enhance interpretability. For instance, Figure 4 clearly illustrates how spatial and temporal bottlenecks affect performance, depending on graph structure and filter size.
4. The paper provides practical insights, such as the effectiveness of temporal row normalization and dilated convolutions in reducing temporal over-squashing supported by both theoretical and empirical evidence.

### Weakness
1. Some figures (such as Figure 2) present heatmaps with specialized axes (e.g., “distance from $t$”), but the paper could better guide the reader through their interpretation. More detailed explanations of how the displayed values relate to over-squashing severity would improve accessibility for a broader audience.
2. While the theoretical equivalence of TTS and T&S architectures in terms of over-squashing is well established, the paper could benefit from a deeper discussion of practical trade-offs, such as optimization stability or data modality that might still make one paradigm preferable in real-world applications.
3. The analysis focuses entirely on factorized convolutional architectures. Extending or comparing the findings to other popular spatiotemporal models, such as recurrent or Transformer-based architectures [1,2,3,4], would make the study more comprehensive. This is not strictly necessary, but would meaningfully strengthen the paper’s overall contribution.

[1] Liu, Kay, et al. "Tgtod: A global temporal graph transformer for outlier detection at scale." Pacific-Asia Conference on Knowledge Discovery and Data Mining. Springer, Singapore, 2025.

[2] Wu, Liming, et al. "Equivariant spatio-temporal attentive graph networks to simulate physical dynamics." Advances in Neural Information Processing Systems 36 (2023): 45360-45380.

[3] Xu, Mingxing, et al. "Spatial-temporal transformer networks for traffic flow forecasting." arXiv preprint arXiv:2001.02908 (2020).

[4] Zhang, Tong, et al. "Spatial–temporal recurrent neural network for emotion recognition." IEEE transactions on cybernetics 49.3 (2018): 839-847.

---

> ### Author Rebuttal · Authors · 2025-07-31
>
> We thank the reviewer for their thoughtful and constructive feedback. We are pleased that the reviewer finds our theoretical analysis rigorous and our empirical results insightful. We respond to the weaknesses raised point-by-point below:
>
> * **W1. Interpretation of heatmaps (Figure 2)**
>   We appreciate this suggestion. Figure 2 displays heatmaps of the normalized entries of the temporal topology matrices. These values correspond to the right-hand side of the upper bound in Theorem 4.1. A lower value for the $(i,j)$ entry indicates a higher risk of temporal over-squashing from time step $t-i$ to $t-j$, as it implies a weaker effective connectivity between the two time steps at the corresponding layer. We have revised the caption of Figure 2 to explicitly clarify this interpretation and added a brief explanatory sentence in the main text to guide readers through the heatmap axes and color scale.
>
> * **W2. Practical trade-offs between TTS and T\&S architectures**
>   Thank you for this insightful suggestion. In the manuscript (Lines 163–174), we briefly highlight the efficiency advantages of TTS, particularly for distributed implementations where temporal processing can be conducted individually across the time series before spatial processing. We agree that a more detailed discussion of practical trade-offs adds value. Accordingly, we have expanded the text to highlight that T\&S architectures may be more suitable when the graph topology varies over time, with each time step potentially associated with a different adjacency matrix. In such cases, TTS models require additional mechanisms to aggregate or align spatial topologies across time steps. This addition complements the analysis of computational complexity for both approaches, as detailed in Appendix C.
>
> * **W3. Scope limited to factorized convolutional architectures**
>   We fully agree that analyzing over-squashing in other STGNN paradigms – such as recurrent or Transformer-based models – is a valuable direction. We acknowledge this limitation in the paper and emphasize that our goal was to provide a formalization of spatiotemporal over-squashing and a deeper analysis in a specific and popular class of STGNNs. Different paradigms (e.g., attention-based or recurrent models) rely on fundamentally different information propagation mechanisms, making it challenging to analyze them under a unified framework. Doing so with comparable theoretical depth would substantially increase the scope and length of the paper and require a level of detail beyond what is feasible within the limits of a conference-length submission. We view our work as laying the theoretical foundation for future explorations into over-squashing in other architectures, and have clarified this in the conclusion section.

---

> > ### Comment · Reviewer_7S5V · 2025-08-05
> >
> > I thank the authors for their thorough and thoughtful responses, which have addressed my concerns. While the current work focuses on convolutional architectures, I encourage the authors to explore extensions to other architectures (e.g., Transformer-based approaches) in future research, as this could further broaden the impact of their contributions.  Overall, I maintain my positive score.

---

> > > ### Author Response · Authors · 2025-08-07
> > >
> > > We thank the reviewer for their time, the positive assessment, and helpful suggestions. We agree that extending our work to a broader range of architectures is a promising direction and plan to explore it in future research.

---

### Comment · Area_Chair_KX7M · 2025-08-05
**Please participate in the discussions and respond to the authors**

Dear Reviewers,

Thank you for your valuable reviews. With the Reviewer-Author Discussions deadline approaching, please take a moment to read the authors' rebuttal and the other reviewers' feedback, and participate in the discussions and respond to the authors. Finally, be sure to complete the "Final Justification" text box and update your "Rating" as needed. Your contribution is greatly appreciated.

Thanks.\
AC

---

### Note · Authors · 2025-08-14

Dear ACs and Reviewers,

we sincerely thank you for your valuable feedback and the time you have dedicated throughout the review and discussion process. We are grateful that reviewers have noted the novelty, significance, and robustness of the theoretical contribution, and that both synthetic and real-world experiments support the theory.

In response to the points raised during the reviewing process, we have incorporated the following planned changes in the revised manuscript:

* Added results combining temporal and graph rewiring to assess their joint effect as mitigation strategies.
* Introduced the *TemporalNeighboursMatch* problem as an additional synthetic task and included corresponding experimental results.
* Improved heatmaps (Figures 2 and 4) by refining captions and enhancing representation and visual descriptions.
* Included an illustrative example demonstrating sensitivity in TCNs.
* Clarified the scope of the work as a theoretical foundation for spatiotemporal over-squashing and expanded the discussion on joint space-time filters and recurrent approaches.
* Expanded the discussion on trade-offs between TTS and T&S, the sink effect in TCNs, and limitations of the proposed mitigation strategies.
* Added a note on the limitation of the upper-bound analysis in the limitations section.

We believe these changes further improve the soundness, clarity, and completeness of our work. Thank you again for your constructive engagement, which we are confident has strengthened the paper.

---

### Decision · Program_Chairs · 2025-09-17

**Decision:**

Accept (poster)

**Comment:**

Summary:
This paper investigates the over-squashing issue in spatiotemporal graph neural networks, a problem where distant nodes fail to effectively exchange information. The authors highlight that the temporal dimension in spatiotemporal graphs amplifies this challenge by increasing the amount of information that needs to be propagated. In this work, the authors formally define the spatiotemporal over-squashing problem and demonstrate its unique characteristics compared to the static case. They validate their findings on both synthetic and real-world datasets, providing insights into their operational dynamics and principled guidance for more effective STGNN designs.

Strengths:
1. Over-squashing in spatiotemporal networks is a critical and underexplored problem.
2. The theoretical framework is robust, and its claims are strongly supported by comprehensive empirical evaluations on both synthetic and real-world datasets.
3. The paper is well-written and easy to follow.


Weaknesses:
1. Some parts of the paper, such as Figure 2, require a more detailed illustration.
2. The analysis primarily focuses on "factorized convolutional approaches" (MPTCNs). By explicitly excluding recurrent architectures and models with joint space-time filters, the work limits the generalizability of its theoretical findings to the broader STGNN landscape.
3. The core ideas of over-squashing, sensitivity analysis via Jacobians, and graph rewiring are established concepts in the GNN literature, which somewhat restricts the originality of the work.


This paper addresses the important and underexplored problem of over-squashing in spatiotemporal graph neural networks. While the task is significant, some of its core ideas are established concepts in the existing GNN literature. I recommend that the authors address the issues suggested by the reviewers in their revised version.